# DE-HALLUCINATING CLIP EMBEDDINGS TO IMPROVE BRAIN-VISION MAPPING

## ABSTRACT

Recent advances in vision-language models, such as CLIP, have enabled their widespread use in brain encoding and decoding, where global image embeddings serve as anchors linking visual stimuli to voxel-level brain responses. However, we observe that CLIP's global visual embeddings often exhibit hallucinatory semantics: they encode objects not explicitly present in an image but inferred from prior associations. This imaginative bias poses a significant challenge for brain-vision mapping, particularly for natural scenes containing multiple annotated objects, where human neural responses are constrained to what is actually perceived. To address this issue, we propose a framework that suppresses CLIP's visual hallucination by integrating object- and concept-level representations. First, we extract object-centric embeddings using segmentation masks, isolating visual features tied to explicitly present objects. Next, we stabilize these diverse segment embeddings with a concept bank of text-derived CLIP embeddings, aligning bottom-up perception with top-down categorical knowledge through cross-attention. The resulting concept-stabilized object features act as corrective signals to be fused with global scene embeddings to form de-hallucinated visual representations. Finally, these representations are used for voxel-wise regression. Experiments on the NSD dataset demonstrate that our method generates representations that better align with category-selective brain regions (bodies, faces, food, places, and words), leading to more accurate and reliable neuro-based image generation compared to standard CLIP regression. These results highlight the importance of suppressing model imagination in bridging human perception with multimodal foundation models and offer a new direction for robust, biologically grounded brain-vision alignment.

## 1 INTRODUCTION

Understanding how the human brain represents visual information remains a central challenge in both neuroscience and machine learning (Mathis et al., 2024). Recent advances in multimodal foundation models, such as CLIP (Radford et al., 2021), have enabled researchers to directly map brain activity onto powerful image-text embedding spaces, offering a promising framework for neural decoding and the study of brain-vision alignment (Cerdas et al., 2025; Yin et al., 2025). A widely adopted approach in this line of work is to anchor voxel-level brain responses to the global image embedding produced by the final layer of CLIP's visual encoder (Wang et al., 2024; Li et al., 2024; Ma et al., 2025; Zhou et al., 2024) [1]. This strategy assumes that the global embedding effectively captures the visual content experienced by a subject, providing a natural bridge between stimulus and brain activity.

However, this assumption overlooks an important limitation. The CLIP model and its embeddings are optimized for open-vocabulary recognition and image-text alignment, but not for faithfully identifying and disentangling multiple objects in complex natural scenes. This shortcoming is especially critical for the Natural Scenes Dataset (NSD) (Allen et al., 2022), where visual stimuli are drawn from MS-COCO (Lin et al., 2014) images containing rich multi-object compositions that trigger corresponding functional Magnetic Resonance Imaging (fMRI) brain responses (Matthews & Jezzard, 2004). For brain-vision mapping, where accurate alignment between visual content and neural representations is essential, such ambiguity in the embeddings becomes a fundamental obstacle to bridge the two modalities. To validate this, we evaluate CLIP on multi-object images with ground-truth annotations.

---

[1] See Appendix A.1 for more related works.

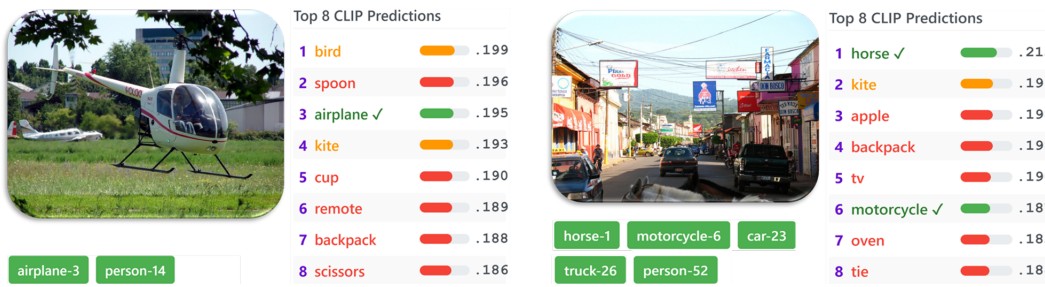

Figure 1: Illustration of CLIP's visual hallucination, where two images display ground-truth objects (green rectangle) and predicted object rankings. Top 8 objects are listed in terms of the similarity scores against canonical prompts: ● means correct prediction, ● shows contextual hallucination (*which fits the context but isn't actually present in the image*), and ● denotes spurious hallucination.

Leveraging CLIP's canonical text prompts ("a photo of an [object]"), we compare the global image embedding against text embeddings of object names. Two systematic problems are observed:

  (i) **Annotated objects frequently fail to appear among the top-ranked**, and;

  (ii) **Spurious, unannotated objects are incorrectly elevated to high ranks.**

Figure 1 exhibits two representative cases. In the first example, the annotated labels are *airplane* and *person*. While CLIP does rank *airplane* third among all 80 MS-COCO object categories, the similarity margin over candidates is negligible. Moreover, the high-ranked predictions fall into two types: (a) reasonable but misleading hallucinations such as *bird* and *kite*, correlating with the contextual cue *sky*; and (b) spurious hallucinations (e.g., *toothbrush*, *apple*, *bowl*), which carry no semantic relation to the image. In more complex cases, the issue worsens. For the second example annotated with *person*, *motorcycle*, *truck*, *horse*, and *car*. CLIP identifies only *horse* and *motorcycle* within its top predictions, while ignoring the rest, replaced by unrelated categories (*apple*, *oven*, etc.).

This evidence suggests that CLIP's global visual embeddings encode imaginative or confounded object information, extending beyond the objects actually present. While tolerable in open-vocabulary recognition tasks, this behavior is fundamentally misaligned with human perception, which exhibits far greater semantic precision, exclusivity, and clarity (Stigliani et al., 2015; Jain et al., 2023; Firestone, 2020). Consequently, directly mapping CLIP's global visual features to brain signals risks introducing hallucinated or diluted semantics, undermining the fidelity and quality of brain-vision alignment.

To mitigate this observed hallucination issue, we propose a multi-step de-hallucination framework that starts with CLIP's global image-level embedding and progressively refines it through two key stages. First, we extract **object-centric embeddings** from segmented image regions, providing a grounded representation of the actual visual content. Next, these object embeddings are anchored to a stable, text-derived concept bank via cross-attention, producing **concept-enriched representations** that ensure semantic consistency and categorical correctness. The enriched object features are then fused with the global embedding to suppress its imaginative bias and form a human-aligned visual representation that balances holistic scene context with grounded concept-corrected object information. Finally, this de-hallucinated representation is mapped to voxel responses through a regression model, yielding a faithful and robust brain-vision correspondences. Our contributions can be summarized into four folds as follows:

- We identify and characterize a hallucination phenomenon in CLIP's visual embeddings within complex scenes and further discuss their potential to undermine brain-vision tasks–a significant concern given CLIP's prominence as a leading model in this domain.

- We propose a novel de-hallucination framework that grounds global embeddings with object- and concept-level refinements, directly mitigating CLIP's tendency to over-imagine.

- Through extensive experiments, we demonstrate that our framework produces semantically faithful brain-to-vision mappings, yielding stronger and more selective activations in category-relevant voxel groups and improving both neural alignment and semantic generation quality.

- We show that our method preferentially enhances the connection to semantically selective voxels, validating its neuroscientific plausibility.

## 2 METHODS

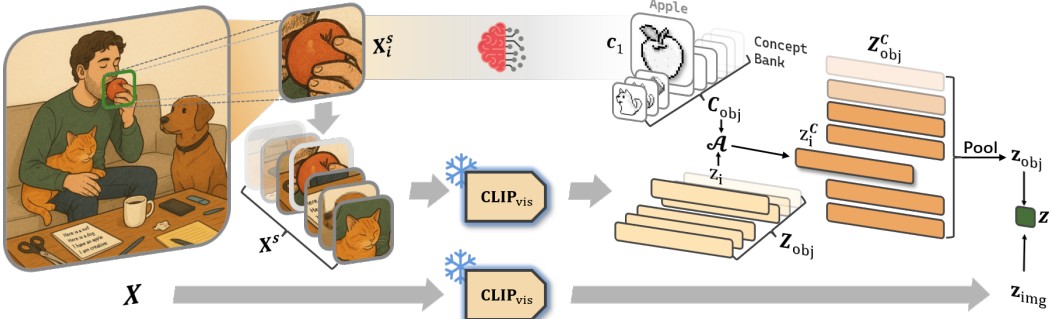

Figure 2: Overview of our de-hallucinated encoding framework: An input visual stimulus $\mathbf{X}$ is first decomposed into multiple object instances $\mathbf{X}^s$. A frozen CLIP Vision Encoder extracts: (1) the class token as a global image embedding $\mathbf{z}_{img}$, and (2) object-level embeddings $\mathbf{Z}_{obj}$ for each segment. A fixed Concept Bank $\mathbf{C}_{obj}$, constructed from CLIP text prompts, provides stable semantic references. Object features $\mathbf{Z}_{obj}$ interact with these concept embeddings via a trainable cross-attention module, yielding concept-aligned object features $\mathbf{Z}_{obj}^{\mathbf{C}}$. Finally, the refined object–concept representations are pooled and fused with the global embedding to produce the de-hallucinated visual feature $\mathbf{z}$, capturing both precise object semantics and holistic scene context.

### 2.1 PROBLEM FORMULATION

Let $\mathcal{D} := \{(\mathbf{X}_j, \mathbf{v}_j)\}_{j=1}^{M}$ denote a neuro-vision dataset, where each pair consists of a visual stimulus and its corresponding neural response. The stimulus $\mathbf{X} \in \mathbb{R}^{H \times W \times 3}$ is an RGB image with a spatial resolution of $H$ and $W$. The neural response $\mathbf{v} \in \mathbb{R}^L$ is a one-dimensional vector of voxel intensities from an fMRI volume, recorded while a subject viewed $\mathbf{X}$. The length $L$ of neural response vector varies across subjects (ranging from 12,682 to 17,907) due to neuro-anatomical differences. As the proposed model is trained and evaluated independently on a per-subject basis, this variation in $L$ does not affect the formulation and can be treated as a fixed value for any given subject.

The goal is to build a subject-specific mapping $\mathcal{M}$ from visual stimuli to corresponding brain responses. Formally, the problem is formulated by solving a regression task and can be described as:

$$\mathcal{M} : \mathbb{R}^{H \times W \times 3} \to \mathbb{R}^L. \tag{1}$$

### 2.2 FRAMEWORK OVERVIEW

Pre-trained CLIP visual features, often trusted as the strongest encoders for representing visual stimuli (Wang et al., 2021; Conwell et al., 2022; 2023; Wang et al., 2023), are in fact prone to hallucinations and can fail to reliably capture the true visual content as pointed out previously in Figure 1.

For this reason, we propose a biologically informed CLIP embedding de-hallucination framework that incrementally refines visual representations before mapping them to neural responses. Following the prior work (Luo et al., 2023a; Cerdas et al., 2025), we begin by extracting a normalized global image-level embedding $\mathbf{z}_{img} \in \mathbb{R}^d$ of the full image $\mathbf{X}$ using the visual encoder of a frozen CLIP model. $d$ here refers to the latent dimension of CLIP space after image-text alignment:

$$\mathbf{z}_{img} := \frac{\text{CLIP}_{vis}(\mathbf{X})}{||\text{CLIP}_{vis}(\mathbf{X})||_2} \in \mathbb{R}^d. \tag{2}$$

While $\mathbf{z}_{img}$ captures high-level semantic context and has proven effective in prior decoding studies, it also inherits biases from CLIP's internet-scale training corpus. Specifically, it can over-weight dominant scene elements or encode "imaginative" semantics not present in the image, leading to mismatches with fMRI signals that reflect concrete perceptual content.

To address these issues, we introduce a two-stage framework for robust, human-aligned visual representations (Figure 2). First, object-centric embeddings capture segmented image objects, grounding the content. Then, these embeddings are enriched and stabilized via alignment with a text-derived concept bank, and eventually fused with the global image representation. This produces a scene-aware and concept-corrected feature, which is mapped to voxel responses through regression.

## 2.3 Object-centric Representation

As mentioned previously, each visual stimulus $\mathbf{X}$ comes from MS-COCO dataset, where images were collected in natural scenes containing multiple objects simultaneously. In such settings, relying solely on global scene embeddings risks conflating distinct object representations and overemphasizing dominant scene elements. Fortunately, MS-COCO provides ground-truth object-level segmentations, allowing us to explicitly disentangle object-level information. Formally, each image can be decomposed as $\mathbf{X} = \mathbf{X}_1^s \cup \mathbf{X}_2^s \cup \cdots \cup \mathbf{X}_N^s$, where $N$ denotes the total number of segmented objects.

To extract object-specific features, we crop each segment $\mathbf{X}_i^s$ from $\mathbf{X}$ and independently pass it through the same CLIP visual encoder as in equation 2, yielding the representation $\mathbf{z}_i$ for $i$-th object:

$$\mathbf{z}_i := \frac{\text{CLIP}_{\text{vis}}(\mathbf{X}_i^s)}{||\text{CLIP}_{\text{vis}}(\mathbf{X}_i^s)||_2} \in \mathbb{R}^d. \tag{3}$$

Collecting all $N$ object-centric embeddings, we obtain a object-level representation $\mathbf{Z}_{\text{obj}}$ that contains fine-grained details for the whole image $\mathbf{X}$:

$$\mathbf{Z}_{\text{obj}} = (\mathbf{z}_1, \mathbf{z}_2, \ldots, \mathbf{z}_N) \in \mathbb{R}^{N \times d}. \tag{4}$$

By isolating individual object segments from cluttered scenes, this representation $\mathbf{Z}_{\text{obj}}$ offers an opportunity to prevent global image-level features $\mathbf{z}_{\text{img}}$ in equation 2 from "hallucinating" absent semantics and instead grounds the representation in human-annotated object categories. Such de-hallucination is particularly advantageous for neural decoding, as it aligns with evidence that distinct voxel populations exhibit selectivity for specific object categories (Stigliani et al., 2015; Jain et al., 2023), thereby providing a closer match to the brain's object-centric encoding of natural vision.

## 2.4 Concept-enriched Representation

While object-level features $\mathbf{Z}_{\text{obj}}$ provide explicit embeddings for segmented object instances present in the image, they remain tied to the raw pixel appearance of individual segments and can still inherit CLIP visual encoder's biases. Such segment-level embeddings may be noisy and diverse due to imperfect boundaries, variable lighting, or occlusion, and they often lack the higher-level semantic stability that guides human perception. Neuroscience research has shown that object recognition in the ventral visual stream involves not only encoding raw visual input but also integrating categorical knowledge, allowing robust recognition across changes in size, pose, or context (Kravitz et al., 2013; Bracci et al., 2017; Henderson et al., 2025; Bougou et al., 2024; Schmidt et al., 2025). This motivates us to enrich object embeddings with stable and semantically grounded conceptual representations.

To this end, we construct a **concept bank** $\mathbf{C}_{\text{obj}} \in \mathbb{R}^{K \times d}$ consisting of $K = 80$ object categories defined by MS-COCO. Each category is represented by a CLIP text embedding using the prompt, "a cropped centered photo of [object]", which aligns with the nature of each segment at pixel-level and provides a semantically consistent and language-aligned reference for each segmented object:

$$\mathbf{C}_{\text{obj}} = (\mathbf{c}_1, \mathbf{c}_2, \ldots, \mathbf{c}_K) \in \mathbb{R}^{K \times d}. \tag{5}$$

These text-derived concept representations are invariant to visual idiosyncrasies of individual images and thus serve as an anchor for stabilizing highly-diverse object segment embeddings. We then integrate the segment-level features $\mathbf{Z}_{\text{obj}}$ with the built concept bank $\mathbf{C}_{\text{obj}}$ using a cross-attention module $\mathcal{A} : \mathbb{R}^{N \times d} \times \mathbb{R}^{K \times d} \times \mathbb{R}^{K \times d} \to \mathbb{R}^{N \times d}$ as shown below:

$$\mathbf{Z}_{\text{obj}}^{\mathbf{C}} = \mathcal{A}(\text{Query} = \mathbf{Z}_{\text{obj}}, \text{Key} = \mathbf{C}_{\text{obj}}, \text{Value} = \mathbf{C}_{\text{obj}}) \in \mathbb{R}^{N \times d}. \tag{6}$$

In the formulation above, the object features $\mathbf{Z}_{\text{obj}} \in \mathbb{R}^{N \times d}$ derived from direct visual stimuli serve as **queries**, which are compared against the **keys** $\mathbf{C}_{\text{obj}} \in \mathbb{R}^{K \times d}$ representing stable human-defined concepts. This matching process establishes correspondences between diverse visual segments and consistent semantic categories, after which the corresponding concept embeddings (the **values**) are retrieved and adjusted accordingly. This design can be viewed as a computational analogue of how the brain integrates perceptual features $\mathbf{Z}_{\text{obj}}$ with category-level knowledge $\mathbf{C}_{\text{obj}}$ to achieve stable and

selective understandings $\mathbf{Z}_{\text{obj}}^{\mathbf{C}} = (\mathbf{z}_1^{\mathbf{C}}, \mathbf{z}_2^{\mathbf{C}}, \ldots, \mathbf{z}_N^{\mathbf{C}}) \in \mathbb{R}^{N \times d}$ for all $N$ perceived objects. Eventually, we pool the attended representations to obtain a stabilized object embedding at conceptual level:

$$\mathbf{z}_{\text{obj}} = \mathsf{AvgPool}(\mathbf{Z}_{\text{obj}}^{\mathbf{C}}) = \frac{1}{N} \sum_{i=1}^{N} \mathbf{z}_i^{\mathbf{C}} \in \mathbb{R}^d, \tag{7}$$

which eventually integrates **bottom-up** visual evidence with **top-down** human-annotated concepts, leading to a stabilized object-level semantics.

## 2.5 FUSED VISUAL REPRESENTATION

We construct the final representation $\mathbf{z} \in \mathbb{R}^d$ by simply fusing the image-level embedding $\mathbf{z}_{\text{img}} \in \mathbb{R}^d$ from equation 2 with the concept-enriched object-level embedding $\mathbf{z}_{\text{obj}} \in \mathbb{R}^d$ from equation 7:

$$\mathbf{z} = \frac{\mathbf{z}_{\text{img}} + \mathbf{z}_{\text{obj}}}{2} \in \mathbb{R}^d. \tag{8}$$

This fusion balances two complementary information streams. First, the global image-level embedding captures holistic scene layout and contextual relations, while the concept-stabilized object-level embedding grounds the representation in explicitly present and human-annotated semantics. Importantly, $\mathbf{z}_{\text{obj}}$ serves as **a corrective signal** that counteracts the tendency of $\mathbf{z}_{\text{img}}$ to overweight dominant scene elements or encode "imagined" semantics not present in the actual stimulus. As a result, the fused representation therefore reflects both the broad visual layout and the precise, semantically anchored content that human participants are likely to perceive.

From a neuroscientific perspective, this integration mirrors how the brain combines coarse global context with fine-grained object-specific information to form robust visual representations across multiple regions of the ventral stream (Kravitz et al., 2013). The resulting feature $\mathbf{z}$ is thus not only semantically richer but also more aligned with voxel-level neural responses.

Finally, the representation $\mathbf{z}$ is mapped to voxel responses using a single layer linear regression model $f : \mathbb{R}^d \to \mathbb{R}^L$ with $L$ denoting the length of fMRI signals:

$$\hat{\mathbf{v}} = f(\mathbf{z}) = \mathbf{W}\mathbf{z} + \mathbf{b} \in \mathbb{R}^L, \tag{9}$$

where $\mathbf{W} \in \mathbb{R}^{L \times d}$ and $\mathbf{b} \in \mathbb{R}^L$ are learnable parameters optimized via a mean squared error (MSE) loss $\mathcal{L}$ against the ground-truth voxel activations $\mathbf{v}$ paired with its visual stimulus $\mathbf{X}$:

$$\mathcal{L} := \mathcal{L}_{\text{MSE}}(\mathbf{v}, \hat{\mathbf{v}}). \tag{10}$$

This deliberately simple regression transparentizes the contribution of the proposed de-hallucination strategy, ensuring that improvements in regression quality arise from $\mathbf{z}$ rather than model complexity.

## 3 EXPERIMENTS

### 3.1 EXPERIMENTAL SETUP

**Dataset**. We evaluate our framework on the Natural Scenes Dataset (NSD) (Allen et al., 2022), the largest fMRI dataset containing responses from eight participants viewing naturalistic images sampled from MS-COCO (Lin et al., 2014). Each participant was intended to complete 40 scanning sessions (10,000 images, each viewed for 3 seconds with three repetitions). Among these stimuli, 1,000 images are shared across all participants and are used as the test set, while the remaining $\sim 9,000$ subject-unique images serve as the training set. Following standard NSD protocols (Gu et al., 2024; Takagi & Nishimoto, 2023; Scotti et al., 2024a;b), we average neural responses across repetitions for each image and conduct voxel-wise standardization. Although the NSD experimental design specifies 40 sessions per subject, the actual number of publicly available sessions varies due to session interruptions and other data-quality issues Allen et al. (2022). As a result, only four subjects (subj01, subj02, subj05, and subj07) have the full 40 sessions. For subj03 and subj06, 32 usable sessions are released, while subj04 and subj08 each contain merely 30 sessions. All models in this paper are trained independently for each subject using the available sessions for that individual.

**Model Training**. Our model focuses on refining CLIP-derived visual embeddings without heavy computation. The cross-attention module is implemented with trainable K/Q/V projections with a

single attention head. The predictive model is implemented as a one-layer linear regression head with a bias term that maps the de-hallucinated CLIP features to voxel activations. All parameters are optimized with a weight decay of 0.05, making the predictor equivalent to ridge regression. We take OpenAI's *clip-vit-large-patch14* as our pretrained and frozen backbone, which fixes the latent feature dimension at $d = 768$. The whole model is trained for 200 epochs, including 10 warm-up epochs. The learning rate is initialized at $7.5 \times 10^{-3}$, annealed gradually to zero with a cosine scheduler. Optimization is performed with AdamW, gradient clipping at $0.8$, and a batch size of $1,000$.

**Image Generation**. To evaluate the effectiveness of our refined features and learned brain-vision mapping, we perform semantic-selectivity image generation following Luo et al. (2023a). Given a trained brain encoder and a subset of voxels $S$ from the whole $L$ voxels, we generate images that maximize the mean activation of voxels in $S$ using a frozen but modified latent diffusion model (Rombach et al., 2022). We employ *stable-diffusion-2-1-base* with $\epsilon$-prediction to generate $512 \times 512$ images. Sampling is performed via multi-step second order DPM-Solver++ (Lu et al., 2025) with 50 steps and an SAG (Hong et al., 2023) of 0.75. Generated images are re-encoded (by *clip-vit-large-patch14* as well) before being passed to the brain encoder of interest. We use a NULL prompt, ensuring the diffusion process is guided solely by voxel activations rather than text conditioning.

All hyperparameters are kept consistent across subjects unless otherwise specified. Experiments are run on two NVIDIA RTX 6000 Ada GPUs. Training requires about 15 minutes per subject, while generation takes 2.5 hours per subject for 1,000 samples.

## 3.2 TARGETED IMPROVEMENTS IN SEMANTICALLY SELECTIVE VOXELS

To assess regression quality, we compute the coefficient of determination $R^2$ per voxel on the held-out testing set. For a straightforward comparison, we treat BrainDiVE's (Luo et al., 2023a) image-to-voxel regression as our baseline and report the voxel-wise $R^2$ improvement, defined as the $R^2$ difference between our model and the baseline. Figure 3 on the right illustrates the distribution of improvements across voxels for each of eight subject.

The distributions are consistently centered just above zero across all subjects, suggesting that our model matches the baseline's predictive power while providing modest overall gains. This stability highlights that our approach maintains reliable regression performance across diverse subjects without introducing systematic degradation. Importantly, while broad gains are not observed, we note an increase in

Figure 3: $R^2$ improvement for 8 subjects.

the maximum average $R^2$ across subjects, rising from 65.54% for the baseline to 67.04% with our framework. This implies that while widespread improvements are limited, localized enhancements do exist and may be concentrated in some specific subsets of voxels.

To investigate if these gains align with functional specialization, we examine relationship between $R^2$ improvement and voxel-level association with semantic categories, quantified by the t-statistics from the NSD functional Region Of Interest (ROI) maps. For representative ROIs (faces, words, places, and bodies), we plot voxel-wise $R^2$ improvements against their corresponding t-values in Figure 4. Each point denotes a voxel, colored red if our model outperforms the baseline and blue otherwise.

Two consistent patterns are observed. First, red points outnumber blue ones across all four ROIs, demonstrating that a larger proportion of voxels benefit rather than degrade under our framework. Second, the regression lines fitted on **all points** in each scatter plot consistently slope upward, indicating that $R^2$ **improvements systematically increase with stronger voxel semantic selectivity**. In other words, our de-hallucination framework directs its gains toward voxels most tightly linked to semantic categories, rather than distributing them uniformly. See more results in Appendix A.7.

These results reinforce the neuroscientific validity of our approach that improvements are not arbitrary but concentrated in semantically meaningful regions of the visual cortex, where human perception is most selective. Thus, our framework not only purifies and stabilizes image features but also enhances the fidelity of brain-vision mapping precisely where neural representations are most robust.

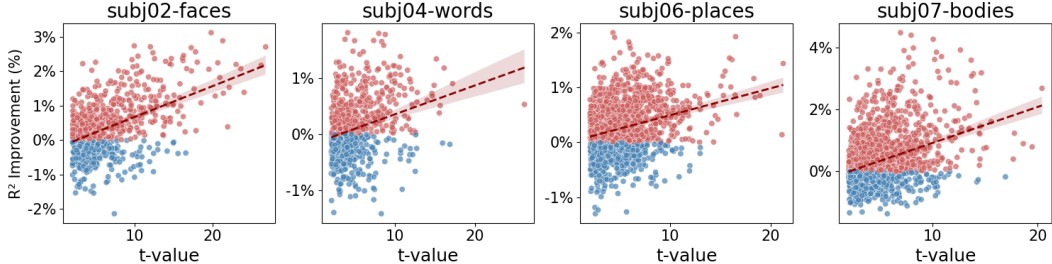

Figure 4: Voxel-wise $R^2$ improvement versus semantic selectivity (t-values) across representative ROIs. Red points (improvement) dominate over blue (decrease), and the upward regression trends indicate that our model captures gains primarily in voxels most strongly linked to semantic categories.

Table 1: Zero-shot category classification accuracy across subjects and voxel groups.

| Subject | Method | Faces | Places | Bodies | Words | Food | Group$_{avg}$ |
|---|---|---|---|---|---|---|---|
| subj01 | BrainDiVE | 61.5 | 91.8 | 36.6 | 34.0 | 33.0 | 51.4 |
|  | Ours | **66.7** | **92.2** | **42.8** | **38.5** | **37.7** | **55.6** |
| subj02 | BrainDiVE | 50.7 | 92.6 | 44.4 | 44.4 | 29.4 | 52.3 |
|  | Ours | **64.5** | **95.9** | **49.8** | **49.6** | **36.5** | **59.3** |
| subj03 | BrainDiVE | 57.9 | 90.9 | 39.8 | 31.1 | **21.5** | 48.2 |
|  | Ours | **67.1** | 90.9 | **49.0** | **47.8** | 18.8 | **54.7** |
| subj04 | BrainDiVE | 56.5 | **94.2** | 42.5 | 20.8 | 27.9 | 48.4 |
|  | Ours | **64.9** | 93.5 | **49.7** | **31.8** | **31.3** | **54.2** |
| subj05 | BrainDiVE | 52.9 | 92.5 | 62.4 | 37.6 | 27.8 | 54.6 |
|  | Ours | **62.3** | **93.8** | **68.4** | **51.4** | **40.7** | **63.3** |
| subj06 | BrainDiVE | 45.4 | 86.5 | 39.5 | 33.4 | 22.2 | 45.4 |
|  | Ours | **57.6** | **86.6** | **43.5** | **48.1** | **28.8** | **52.9** |
| subj07 | BrainDiVE | 47.2 | 92.5 | 41.1 | 19.8 | 26.7 | 45.5 |
|  | Ours | **62.5** | **92.6** | **46.1** | **25.4** | **27.8** | **50.9** |
| subj08 | BrainDiVE | 46.0 | 82.2 | 30.3 | 16.6 | 17.8 | 38.6 |
|  | Ours | **46.3** | **84.4** | **31.9** | **19.0** | **24.2** | **41.2** |
| subj$_{avg}$ | BrainDiVE | 52.3 | 90.4 | 42.1 | 29.7 | 25.8 | 48.1 |
|  | Ours | **61.5** | **91.2** | **47.7** | **39.0** | **30.7** | **54.0** |

## 3.3 Semantic Consistency of Generated Images

We assess whether our de-hallucinated representations yield a more accurate brain-vision mapping by producing reconstructions that more faithfully reflect the object categories encoded by functional voxel groups defined in neuroscience (Stigliani et al., 2015; Jain et al., 2023), thereby preserving the intended category semantics. Following the Luo et al. (2023a) protocol, we employ the trained brain encoder to guide a diffusion model. For each voxel group (faces, food, bodies, places, and words), we generate 1,000 synthetic images by optimizing diffusion sampling to maximally activate the selected voxels. These images are expected to reflect the internal representations carried by each voxel group.

To establish a comparison, we perform generations with both our proposed model and the standard regression baseline. We then evaluate the semantic fidelity of the generated images using CLIP-based probing. Specifically, we feed each generated image into the same CLIP model and perform zero-shot classification using category-specific prompts (see Appendix A.10 for details). Each image is forced into one of the five categories, and we compute the classification accuracy, i.e., the percentage of images assigned to the voxel group's preferred category. If the generations faithfully capture the semantics encoded in the brain, CLIP should consistently identify them with the correct category.

Table 1 reports the results across eight subjects. Our method achieves consistent improvements in semantic alignment across nearly all voxel groups. For instance, face-selective voxels show notable gains (e.g., $+13.8\%$ for subj02 and $+15.3\%$ for subj07), reflecting the ability of concept-enriched features to capture fine-grained category structure. Similarly, word-selective voxels benefit strongly

Figure 5: Qualitative comparison of generated images for subj01 between BrainDiVE and our method. The rows correspond to five semantic categories (bodies, faces, food, places, and words). For each category, we visualize the top-8 images selected from 1,000 generated samples, ranked by their average predicted activation across the voxels of the respective semantic ROI.

from our correction (+9.3% on average), mitigating a known weakness of global CLIP embeddings in representing abstract or symbolic stimuli (Materzyńska et al., 2022; Bhalla et al., 2024; Abbasi et al., 2025). Food-related voxels also show substantial improvements for most subjects (e.g., +12.9% for subj05), highlighting the value of grounding generations in explicit object concepts. The places category is already highly saturated under BrainDiVE (above 90% accuracy for most subjects), yet our approach still yields marginal but consistent gains or maintains performance.

In addition to quantitative evaluation, we include a qualitative comparison of images generated for subj01 by BrainDiVE and our method (Figure 5). Across all semantic categories, both approaches produce images that broadly match the category preferences of the corresponding ROI, indicating consistent high-level semantics. However, several differences highlight the improved quality of our brain-vision mapping. In the **food** row, the third BrainDiVE sample depicts a non-food object with noticeable distortions, whereas all samples generated by our method are clear and unambiguous food items. Similar patterns are observed in the **words** category. BrainDiVE often outputs shapes or textures only weakly suggestive of textual structure, while our method more reliably produces glyph-like or typographic elements, reflecting stronger alignment with the intended semantic content.

These qualitative patterns reinforce our quantitative findings that incorporating object-grounded features, enriched by stable semantic concepts, helps mitigate the imaginative bias present in global CLIP embeddings. While hallucinations are not fully eliminated, the fused representation shifts the distribution toward more faithful category-level semantics. As a result, the generated images better reflect the underlying brain-represented categories, improving semantic consistency across subjects and voxel groups and supporting a more accurate, biologically grounded brain-vision mapping.

## 3.4 NEURAL ACTIVATION ALIGNMENT

In the previous subsection, we demonstrated that images generated with our brain encoder achieve high semantic fidelity and cross-subject consistency. Here, we assess whether these generations also trigger neural responses that align with the category-specific selectivity of functional voxel groups. Concretely, for each subject and semantic category (faces, places, bodies, words, and food), we feed the generated images from two models into their trained brain encoder respectively and compute the mean predicted activation within the corresponding voxel group. This measure captures how strongly the generations drive the expected neural activations.

As shown in Figure 6, radar plots for three representative subjects (subj03, subj04, and subj05) reveal a consistent pattern that our method yields stronger and more category-selective activations than the BrainDiVE baseline. The gains are particularly pronounced for visually heterogeneous categories, such as bodies and words, where our stabilized concept-enriched features provide a corrective signal that better captures category-specific neural tuning. Check Appendix A.8 for plots on all subjects.

Overall, these results highlight that our de-hallucinated features not only preserve semantic consistency at the perceptual level but also improve the neural plausibility of generations. By mitigating

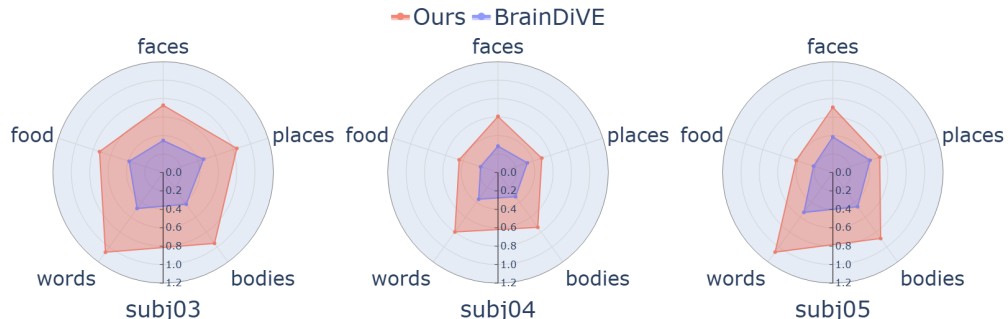

Figure 6: Neural activation alignment across semantic voxel groups. Radar plots show mean predicted activations for five voxel groups (bodies, faces, food, places, and, words) in three representative subjects (subj03, subj04, and subj05). Generations with our method (red) consistently produce stronger and more category-selective activations than the BrainDiVE baseline (blue), particularly for visually variable categories, such as bodies, food, and words.

CLIP's bias, our mapping produces voxel-group activations that more faithfully reflect the brain's true category preferences, establishing a tighter alignment between generated content and brain activity.

## 3.5 CROSS-MODEL VALIDATION

Previous subsections demonstrate that our framework delivers generations with richer semantic fidelity and a tighter correspondence to functional voxel activations. A remaining question, however, is whether these gains stem merely from improved image quality or whether our feature de-hallucination strategy effectively unlocks a more powerful and reliable brain-to-vision mapping compared to the baseline.

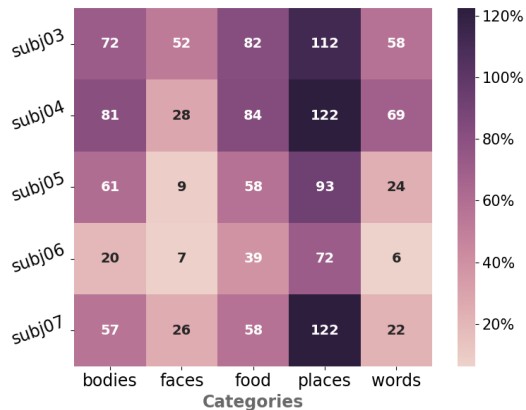

Figure 7: Delta heatmap shows the relative gain in predicted voxel activations (ours-baseline) / baseline when feeding the same generated images.

To figure these effects out, we perform a cross-model validation. Specifically, we feed the images generated by our framework through both our encoder and the baseline encoder, then compare the predicted voxel activations. For each subject and semantic voxel group, we calculate the relative gain in terms of "(ours - baseline) / baseline" and visualize the results as a delta heatmap, highlighting where our approach most effectively enhances the brain-vision mapping. Figure 7 reveals clear and systematic patterns. Across selected five subjects, our model consistently produces stronger activations than the baseline given the same visual inputs. The most significant improvements were observed for *places*, where relative gains range from +72% to +120%. Notably, although zero-shot classification already reaches near-saturated performance on our generated places images (Table 1), our brain encoder is still capable of driving substantially higher activations in place-selective voxels, confirming that our de-hallucination framework establishes a stronger mapping to underlying neural selectivity. More results can be found in Appendix A.9.

The second category showing substantial gains is *food*, which benefits from the availability of 10 annotated object categories within MS-COCO. These annotations enrich food-object representations, enabling our refinement strategy to build stronger visual features that the encoder captures more effectively, leading to consistent activation boosts. In contrast, *faces* and *bodies* exhibit smaller but still positive improvements. We attribute this to the limited availability of human-related object categories–merely *person*–in the label space of training data, leading to marginal refinement allocation across face- and body-related voxels despite their different neural specializations.

At the subject level, subj06 shows the smallest overall activation gains, though still positive across all five semantic groups. This may reflect two factors: (i) distributional differences in the visual stimuli seen by subj06, with fewer overlaps among the five evaluated categories, and (ii) fewer training samples relative to other subjects, which limits the effectiveness of the proposed refinement strategy.

These consistent activation boosts obtained under identical generations indicate that our improvements extend beyond semantic fidelity. Our de-hallucination strategy directly strengthens the brain-vision mapping, refining the feature space to trigger more robust and category-selective neural responses.

## 3.6 ABLATION STUDY

Our proposed de-hallucination framework relies on three levels of representation–image features, object-centric features, and concept-enriched features–together with their refinement and fusion. To understand the contribution of each component, we conduct ablation studies examining how progressively integrating these representations improves the construction of a reliable brain-vision mapping.

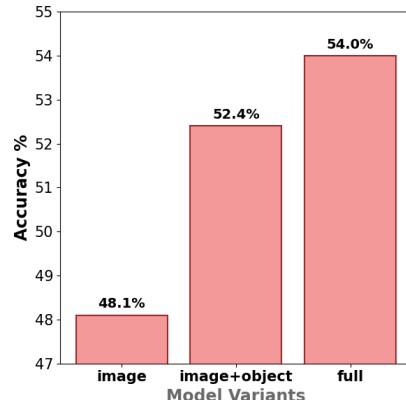

Figure 8: Ablation on features.

**Representational Analysis & Comparison.** We directly compare the original CLIP global embedding with our de-hallucinated embedding using a retrieval-based quantitative evaluation in Appendix A.3. This analysis formalizes the phenomenon illustrated in Figure 1 and provides a dataset-level assessment of how each representation suppresses spurious object hallucinations and prioritizes true objects. We additionally include visual comparisons following the style of Figure 1 in Appendix A.6 to give intuitive examples alongside the quantitative results.

**Semantic Selectivity**. Following the zero-shot semantic selectivity evaluation in Section 3.3, we measure top-1 accuracy across the eight NSD subjects for each ablated variant (Figure 8). The full model–combining image, object, and concept features–achieves the highest accuracy of 54.0%. A mid-level variant that fuses global image features with pooled object-centric features yields intermediate performance, falling between the image-only and full-model results.

These findings show a clear gain from image-level to object-level to concept-level representations. Each stage incrementally mitigates hallucinations and enhances the alignment between visual representations and neural responses. This validates the necessity of all components and highlights how stabilized and semantically grounded features contribute to a high-fidelity brain-vision mapping.

## 4 CONCLUSION

In this work, we identify and characterize the hallucination problem in CLIP's visual embeddings and demonstrate its potential drawbacks for brain-vision applications. To address this issue, we introduce a lightweight yet effective de-hallucination framework that grounds global image embeddings through object- and concept-level refinements, explicitly reducing CLIP's tendency to over-imagine objects. By integrating bottom-up visual evidence with top-down concept priors, our framework stabilizes object-level semantics in a way that mimics human visual processing, thereby enabling a more faithful and biologically plausible brain-vision mapping. The state-of-the-art performance in semantic selectivity tasks and the capability to drive stronger, category-specific neural activations in semantic voxel groups highlight the importance of correcting semantic hallucinations in vision-language models for neuroscience applications, and establish our framework as a principled step toward building a robust and high-quality brain-vision mapping.

In future work, we plan to extend our framework to a multi-subject setting, enabling the mapping to leverage larger datasets and learn a unified representation for brain-vision alignment across individuals. Beyond improving generalization, this extension will also allow us to examine inter-subject variability and uncover shared versus personalized neural representations. We also aim to expand our analysis to additional semantic voxel groups, providing a richer understanding of how diverse brain regions encode object categories. Collaborating closely with neuroscientists, we will further investigate how voxel-level activation patterns relate to semantic object representations. Through these interdisciplinary efforts, we aim to pave the way toward brain-based encoding and decoding models that are both more interpretable and more faithful to neuroscientific mechanisms.

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

# A APPENDIX

## A.1 RELATED WORK

### A.1.1 HALLUCINATION IN VISION LANGUAGE MODELS

Recent research has extensively investigated object hallucination in vision-language models (VLMs). Deng et al. (2024) propose CLIP-Guided Decoding (CGD), a training-free method that leverages CLIP's image-text similarity to select visually grounded sentences during decoding, thereby reducing hallucination. CLIP-DPO (Ouali et al., 2024), which incorporates CLIPScore as a reward within the Direct Preference Optimization framework, grounding generation more closely to visual evidence.

A foundational contribution comes from Li et al. (2023), who systematically study hallucination in LVLMs and establish POPE (Popular Object Prompted Evaluation), a benchmark now widely used for quantifying hallucination severity. Expanding on this, Chen et al. (2024) show that hallucinations are amplified when VLMs must handle multiple objects, as models tend to rely on spurious correlations. Similarly, Favero et al. (2024) introduces M3ID, an instruction-tuning method that explicitly grounds responses in visual input, reducing dependence on language priors.

Other works target architectural and data-level interventions. Yang et al. (2025) identify specific "hallucination heads" within the attention mechanism and propose lightweight interventions, both training-free and fine-tuned, to mitigate them. Zhang et al. (2024) take a data-centric view, augmenting instruction-tuning corpora with reflective pairs that contrast grounded versus hallucinated outputs. Meanwhile, Jiang et al. (2024) propose a comprehensive, multi-dimensional framework to evaluate hallucinations across object, attribute, and relational levels. Finally, Kaul et al. (2024) introduce THRONE, a benchmark focused on free-form generation that evaluates grounding beyond standard VQA tasks.

Previously proposed methods for addressing hallucination in VLMs typically rely on CLIP's image–text similarity scores to suppress spurious concepts. However, our analysis reveals that CLIP's vision encoder alone exhibits hallucination, independent of the text encoder. The work most closely related to ours is Liu et al. (2024), which shows that standard CLIP models often rank hallucinated captions—descriptions of objects absent from the image—higher than correct ones. On their benchmark, CLIP selects the correct caption in only 19% of cases, indicating that hallucination originates primarily from the visual embedding rather than cross-modal alignment. Their approach uses counterfactual data augmentation and fine-tuning to explicitly reduce hallucinations. While consistent with our observation of visual hallucination, their goal differs fundamentally. They aim to train a more hallucination-resistant CLIP, whereas we focus on a training-free procedure to mitigate hallucinated visual content specifically for constructing brain–vision mappings. This strategy allows easy generalization to other vision models. In short, our work tackles a distinct problem—**how CLIP's hallucination affects neural encoding**—and proposes a solution tailored to this setting, rather than general VLM improvement.

### A.1.2 CLIP IN BRAIN-VISION TASKS

Decoding visual information from neural activity is fundamentally a multimodal problem, requiring the integration of at least two modalities: visual inputs and neural recordings (e.g., fMRI). Existing methods for aligning these modalities generally fall into two categories, distinguished by the projection direction between visual and neural representational spaces.

The first line of work maps neural signals into a pre-trained latent visual space. For example, Scotti et al. (2024a) encodes flattened 3D cortical activity patterns into the embedding space of CLIP. Along similar lines, Takagi & Nishimoto (2023) predicts latent representations of viewed images from fMRI signals in the early visual cortex. More recent approaches Wang et al. (2024); Chen et al. (2023); Lin et al. (2022); Scotti et al. (2024b) leverage pre-trained generative models–including GANs (Ratan Murty et al., 2021) and diffusion models(Ferrante et al., 2024)–for reconstruction tasks. These methods benefit from large-scale visual training data and avoid the need to re-train computationally demanding generative architectures, which is impractical given the limited availability of paired neural-visual datasets.

The second line of work reverses this direction by projecting visual features into the neural space. This strategy is particularly useful for constructing synthetic stimuli that selectively activate targeted brain regions, thereby offering insights into feature preferences across cortical areas. Representative studies include Luo et al. (2023a;b); Gu et al. (2022), which examine neural response patterns elicited by feature-driven synthetic stimuli. More recent efforts have further extended this paradigm. For example, Gong et al. (2024) leverages CLIP's semantic space to map continuous fMRI signals onto video keyframes, enabling temporally coherent neural-to-video decoding. Similarly, BrainNRDS (Yeung et al., 2025) decodes visual motion information directly from fMRI activity and uses it to animate static images. In addition, BrainSail (Luo et al., 2024) introduces a fine-grained framework for characterizing how diverse semantic concepts in naturalistic scenes are represented across the human visual cortex.

A.2 DETAILED ZERO-SHOT CLASSIFICATION WITH DIFFERENT LEVELS OF FEATURES

In Table 2 below, letters I, O, and C stands for Image, Object, and Concepts respectively. The classification procedure is the same as Section 3.3 in the main paper.

Table 2: Zero-shot category classification accuracy across subjects and voxel groups with different levels of features.

| Subject | Features | Faces | Places | Bodies | Words | Food | Mean |
|---------|----------|-------|--------|--------|-------|------|------|
| subj01 | I | 61.5 | 91.8 | 36.6 | 34.0 | 33.0 | 51.4 |
| | I+O | **69.3** | **93.2** | 41.8 | 31.3 | 36.9 | 54.5 |
| | I+O+C | 66.7 | 92.2 | **42.8** | **38.5** | **37.7** | **55.6** |
| subj02 | I | 50.7 | 92.6 | 44.4 | 44.4 | 29.4 | 52.3 |
| | I+O | 64.0 | 95.1 | **50.4** | 45.1 | 34.9 | 57.9 |
| | I+O+C | **64.5** | **95.9** | 49.8 | **49.6** | **36.5** | **59.3** |
| subj03 | I | 57.9 | 90.9 | 39.8 | 31.1 | 21.5 | 48.2 |
| | I+O | 61.9 | 89.1 | 42.7 | 37.4 | **22.6** | 50.7 |
| | I+O+C | **67.1** | **90.9** | **49.0** | **47.8** | 18.8 | **54.7** |
| subj04 | I | 56.5 | 94.2 | 42.5 | 20.8 | 27.9 | 48.4 |
| | I+O | **65.1** | **96.2** | 48.6 | 23.0 | **31.5** | 52.9 |
| | I+O+C | 64.9 | 93.5 | **49.7** | **31.8** | 31.3 | **54.2** |
| subj05 | I | 52.9 | 92.5 | 62.4 | 37.6 | 27.8 | 54.6 |
| | I+O | **64.0** | **94.1** | 60.4 | 41.6 | 34.3 | 58.9 |
| | I+O+C | 62.3 | 93.8 | **68.4** | **51.4** | **40.7** | **63.3** |
| subj06 | I | 45.4 | 86.5 | 39.5 | 33.4 | 22.2 | 45.4 |
| | I+O | 55.1 | **91.0** | 39.4 | 42.7 | 23.4 | 50.3 |
| | I+O+C | **57.6** | 86.6 | **43.5** | **48.1** | **28.8** | **52.9** |
| subj07 | I | 47.2 | 92.5 | 41.1 | 19.8 | 26.7 | 45.5 |
| | I+O | 58.2 | **93.1** | **49.9** | **27.1** | **28.5** | **51.4** |
| | I+O+C | **62.5** | 92.6 | 46.1 | 25.4 | 27.8 | 50.9 |
| subj08 | I | 46.0 | 82.2 | 30.3 | 16.6 | 17.8 | 38.6 |
| | I+O | **52.5** | 83.9 | **32.0** | **22.0** | 24.2 | **42.9** |
| | I+O+C | 46.3 | **84.4** | 31.9 | 19.0 | **24.2** | 41.2 |
| average | I | 52.3 | 90.4 | 42.1 | 29.7 | 25.8 | 48.1 |
| | I+O | 61.3 | **92.0** | 45.7 | 33.8 | 29.6 | 52.4 |
| | I+O+C | **61.5** | 91.2 | **47.7** | **39.0** | **30.7** | **54.0** |

A.3 DETAILED REPRESENTATIONAL SIMILARITY ANALYSIS FOR ALL SUBJECTS

In Table 3, the letters I, O, and C denote Image, Object, and Concept, respectively. We evaluate our representational semantics against CLIP's text embeddings in an information retrieval task. The

evaluation employs top-10 metrics: Recall, mean Average Precision (mAP), Area Under the ROC Curve (AUC), and Normalized Discounted Cumulative Gain (nDCG). The combined image and object representations achieve the best performance, validating our design of incorporating segmented objects to generate object-level fine-grained features. Conversely, the full model incorporating image, object, and concept features performs suboptimally. This result is expected and reasonable, as our concept-enriched features are derived from a different objective: they are generated through a cross-attention mechanism between object features and a concept bank, regressed to predict voxel activations. Consequently, these concept features are optimized for brain mapping and are inherently divergent from the semantic space of CLIP's text embeddings by a learnable value embedding matrix, which is the benchmark for this retrieval task.

Table 3: Detailed representational similarity analysis.

| Subject | ID | Image | Object | Concept | Recall | mAP | AUC | nDCG |
|---|---|---|---|---|---|---|---|---|
| subj01 | Full | ✓ | ✓ | ✓ | .673 | .621 | .794 | .717 |
|  | 1 | ✓ |  |  | .735 | .696 | .828 | .779 |
|  | 2 |  | ✓ |  | .796 | .712 | .788 | .708 |
|  | 3 | ✓ | ✓ |  | .822 | .765 | .836 | .783 |
|  | 4 |  | ✓ | ✓ | .115 | .077 | .164 | .108 |
| subj02 | Full | ✓ | ✓ | ✓ | .543 | .544 | .832 | .736 |
|  | 1 | ✓ |  |  | .735 | .696 | .828 | .779 |
|  | 2 |  | ✓ |  | .796 | .712 | .788 | .708 |
|  | 3 | ✓ | ✓ |  | .822 | .765 | .836 | .783 |
|  | 4 |  | ✓ | ✓ | .097 | .055 | .124 | .064 |
| subj03 | Full | ✓ | ✓ | ✓ | .536 | .553 | .854 | .777 |
|  | 1 | ✓ |  |  | .731 | .695 | .831 | .782 |
|  | 2 |  | ✓ |  | .796 | .713 | .791 | .712 |
|  | 3 | ✓ | ✓ |  | .822 | .766 | .835 | .785 |
|  | 4 |  | ✓ | ✓ | .063 | .042 | .080 | .050 |
| subj04 | Full | ✓ | ✓ | ✓ | .515 | .484 | .771 | .654 |
|  | 1 | ✓ |  |  | .730 | .695 | .831 | .782 |
|  | 2 |  | ✓ |  | .795 | .713 | .791 | .712 |
|  | 3 | ✓ | ✓ |  | .822 | .766 | .836 | .786 |
|  | 4 |  | ✓ | ✓ | .064 | .042 | .100 | .061 |
| subj05 | Full | ✓ | ✓ | ✓ | .633 | .581 | .772 | .684 |
|  | 1 | ✓ |  |  | .735 | .696 | .828 | .779 |
|  | 2 |  | ✓ |  | .796 | .712 | .788 | .708 |
|  | 3 | ✓ | ✓ |  | .822 | .765 | .836 | .783 |
|  | 4 |  | ✓ | ✓ | .074 | .040 | .078 | .042 |
| subj06 | Full | ✓ | ✓ | ✓ | .628 | .607 | .805 | .748 |
|  | 1 | ✓ |  |  | .731 | .695 | .831 | .782 |
|  | 2 |  | ✓ |  | .796 | .713 | .791 | .712 |
|  | 3 | ✓ | ✓ |  | .822 | .766 | .835 | .785 |
|  | 4 |  | ✓ | ✓ | .083 | .049 | .111 | .062 |
| subj07 | Full | ✓ | ✓ | ✓ | .588 | .592 | .847 | .781 |
|  | 1 | ✓ |  |  | .735 | .696 | .828 | .779 |
|  | 2 |  | ✓ |  | .796 | .712 | .788 | .708 |
|  | 3 | ✓ | ✓ |  | .822 | .765 | .836 | .783 |
|  | 4 |  | ✓ | ✓ | .078 | .047 | .092 | .061 |
| subj08 | Full | ✓ | ✓ | ✓ | .544 | .500 | .760 | .645 |
|  | 1 | ✓ |  |  | .730 | .695 | .831 | .782 |
|  | 2 |  | ✓ |  | .795 | .713 | .791 | .712 |
|  | 3 | ✓ | ✓ |  | .822 | .766 | .836 | .786 |
|  | 4 |  | ✓ | ✓ | .092 | .066 | .152 | .099 |
| average | Full | ✓ | ✓ | ✓ | .583 | .560 | .804 | .718 |
|  | 1 | ✓ |  |  | .733 | .695 | .830 | .780 |
|  | 2 |  | ✓ |  | .796 | .713 | .789 | .710 |
|  | 3 | ✓ | ✓ |  | **.822** | **.765** | **.836** | **.784** |
|  | 4 |  | ✓ | ✓ | .083 | .052 | .113 | .068 |

## A.4 Hyperparameter Tuning

Table 4: Ablation studies on prompts, heads, and fusion hyperparameters

| ID | $\lambda$ | Concept Prompt | Head(s) | Accuracy (%) |
|---|---|---|---|---|
| Full | 0.5 | a centered cropped photo of [object] | 1 | 42.8 |
| 1 | 0.25 | | | 41.9 |
| 2 | 0.75 | | | **44.3** |
| 3 | 1.00 | | | 36.6 |
| 4 | | a photo of [object] | | 45.3 |
| 5 | | an isolated [object] | | 46.7 |
| 6 | | a detailed view of [object] | | **50.5** |
| 7 | | | 2 | 43.5 |
| 8 | | | 4 | **44.8** |
| 9 | | | 6 | 36.3 |
| 10 | | | 8 | 35.0 |

To better understand the contributions of each design component in our framework, we conducted targeted ablation studies on three key hyperparameters for subj01 on the **bodies** category. The results are summarized in Table 4, where blank cells denote settings identical to those used in the full model.

For **representational fusion**, we reformulate equation 8 as

$$\mathbf{z} = \lambda \mathbf{z}_{\text{img}} + (1 - \lambda)\mathbf{z}_{\text{obj}} \in \mathbb{R}^d, \tag{11}$$

where $\lambda$ controls the balance between image-level and object-level features. Varying $\lambda$ shows that omitting object information leads to a substantial drop in performance to 36.6%, while intermediate fusion strengths remain stable. A slightly higher weight $\lambda$ of 0.75 yields the best performance.

We next ablated the **concept-bank** construction by testing alternative prompt templates. While the default prompt, "a centered cropped photo of [object]," provides a strong baseline (42.8%), more descriptive prompts can yield stronger category-specific alignment. In particular, "a detailed view of [object]" produces a more effective concept bank for this case, increasing accuracy to 50.5%.

Finally, we assessed the number of **attention heads** in the cross-attention module. A single head–our default choice–proves sufficient, and modest increases offer slight improvements (e.g., 4 heads at 44.8%). However, performance degrades considerably as the module becomes overly parameterized, with accuracy dropping sharply beyond 6 heads.

## A.5 FEATURE VISUALIZATION

In Figure 9, we project high-dimensional visual features in shared CLIP space into a 2D plane using UMAP to reveal their semantic relationships. For each test image containing a person, two points are plotted: the original CLIP visual feature (red) and the de-hallucinated feature produced by our framework (green), with a gray line connecting them to trace the feature trajectory. The key reference point is the projected location of the text embedding for the prompt "a photo of a person.

From this visualization, we can draw two primary observations. First, there is a consistent and significant separation between the cluster of original CLIP visual embeddings and the target text concept embedding, indicating a semantic image-text gap. In addition, the connecting lines show a clear and directional convergence that the de-hallucinated features (green) systematically shift from their original CLIP positions (red) toward the concept centroid. This collective movement is not random. It demonstrates that our framework successfully directs the feature representation to reduce spurious correlations and enrich its semantic alignment specifically with the "person" concept, effectively mitigating the representation gap observed in the baseline model.

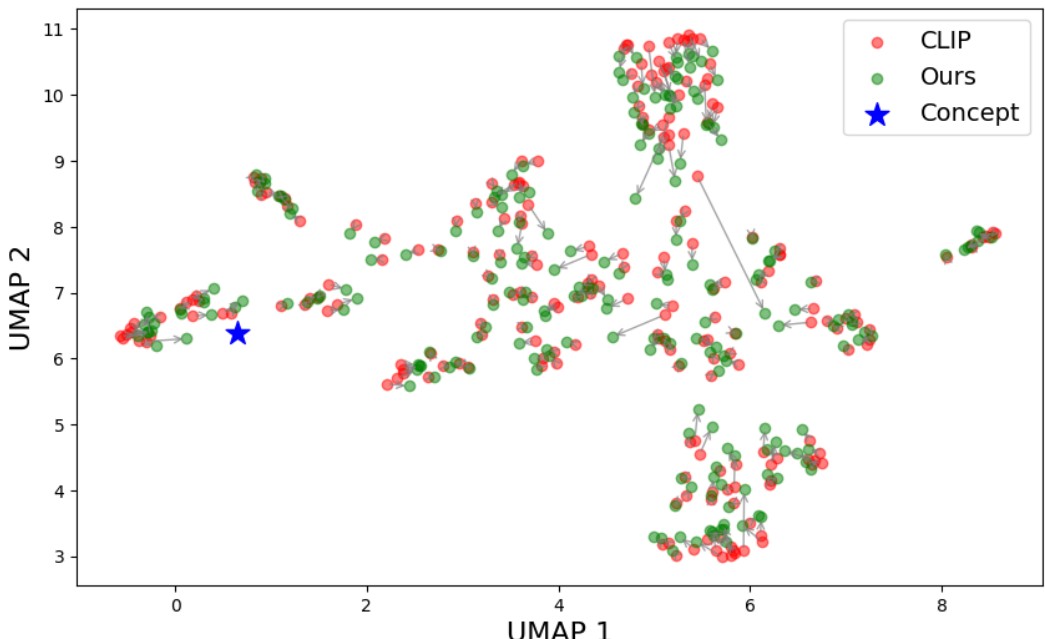

Figure 9: Features Visualization for images contain *person*.

## A.6 VISUAL COMPARISON OF ORIGINAL VS. DE-HALLUCINATED CLIP

Figure 10: Ranking and Similarity Changes 1-4

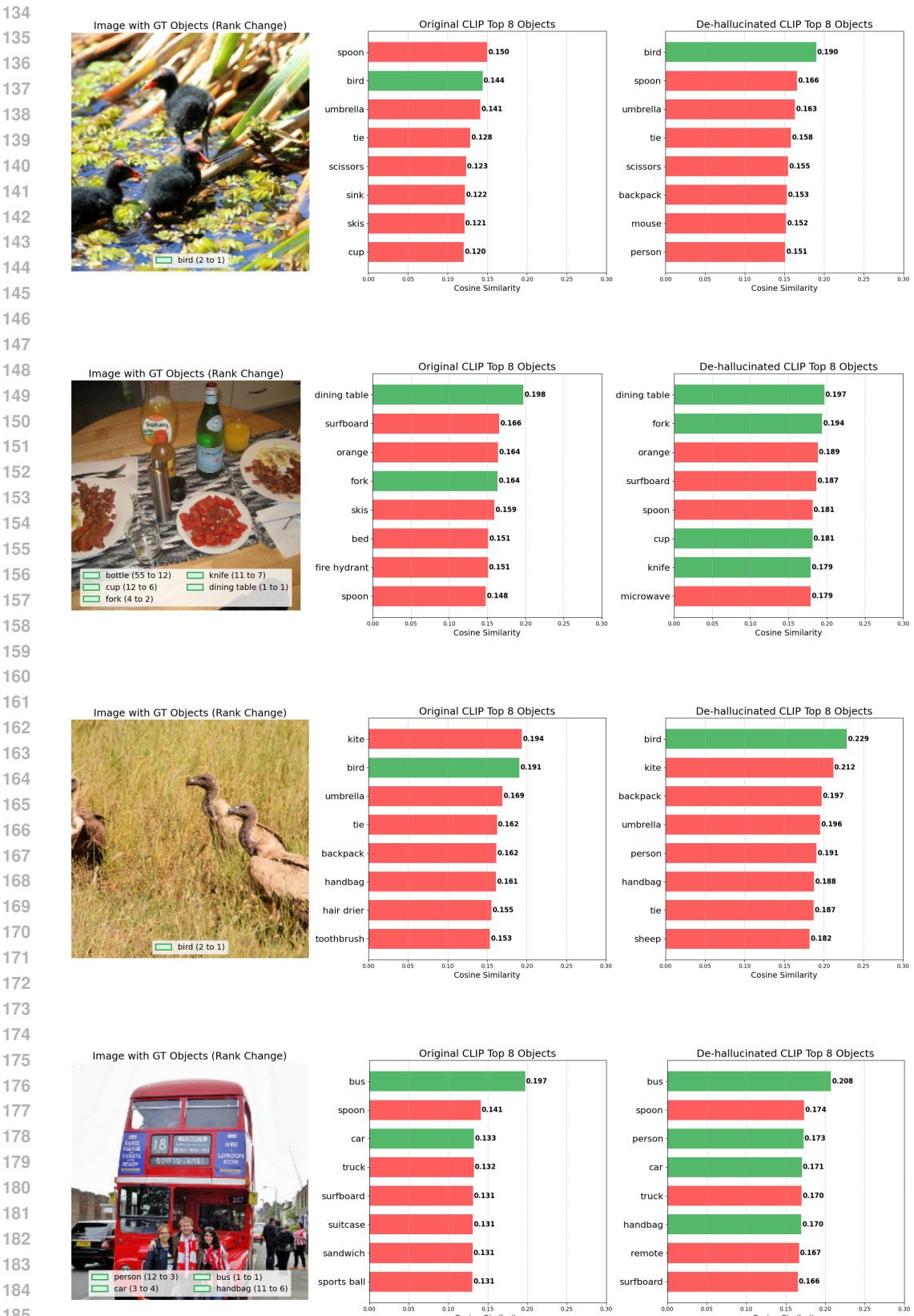

Figure 11: Ranking and Similarity Changes 5-8

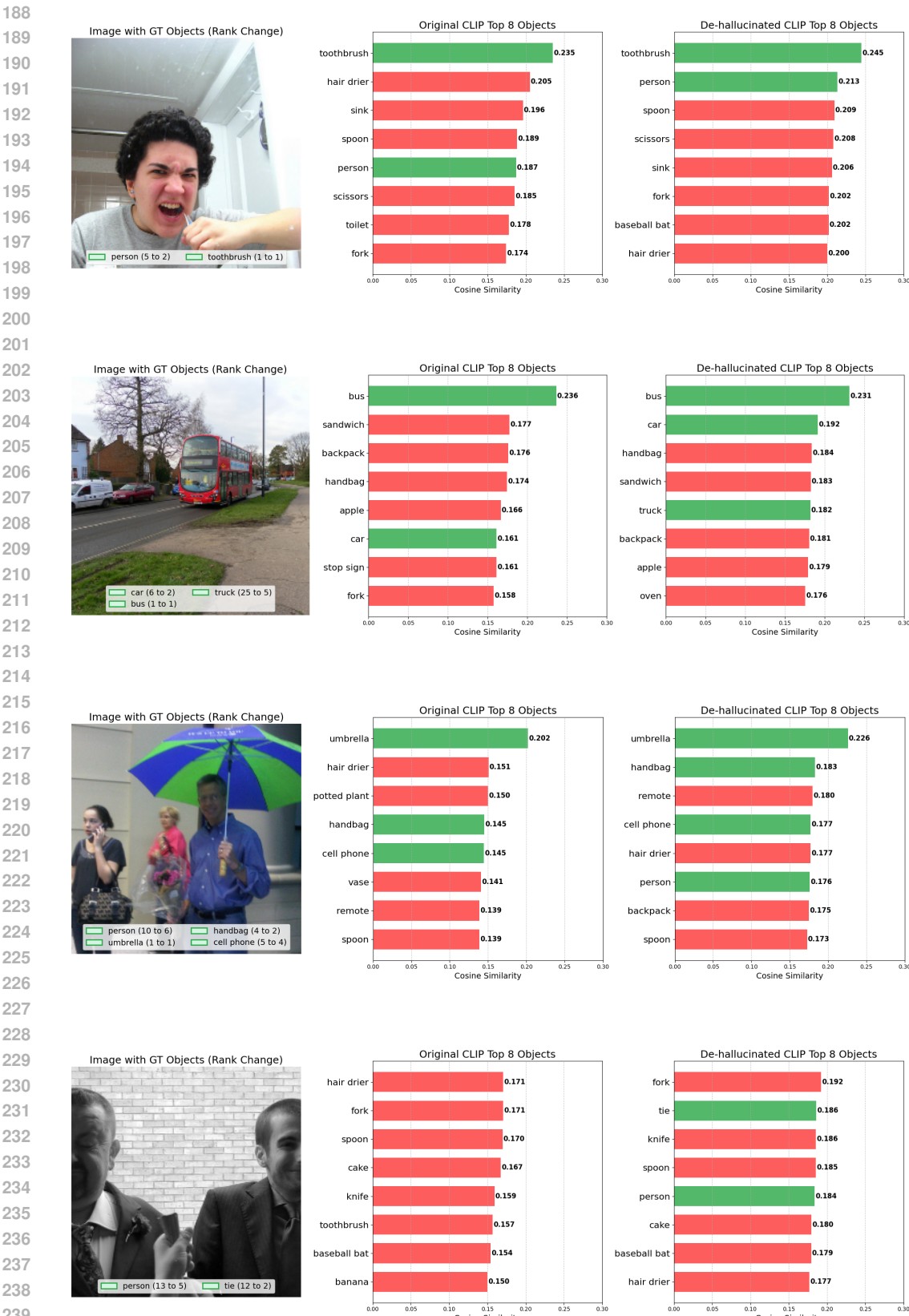

Figure 12: Ranking and Similarity Changes 9-12

## A.7 MORE RESULTS ON VOXEL-WISE IMPROVEMENT

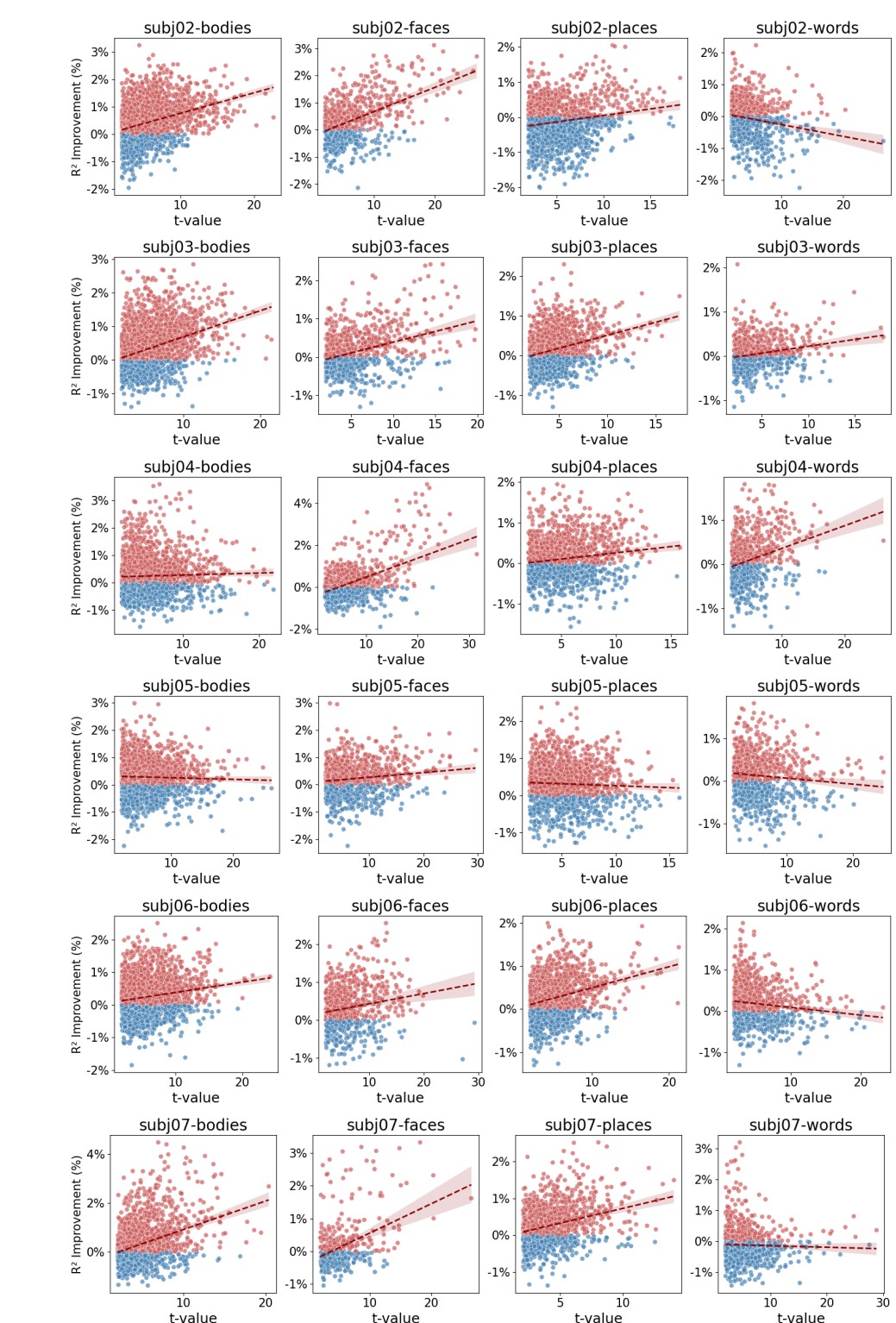

Figure 13: Voxel-wise $R^2$ improvement versus semantic selectivity across representative ROIs.

## A.8 MORE RESULTS ON MEAN NEURAL ACTIVATION

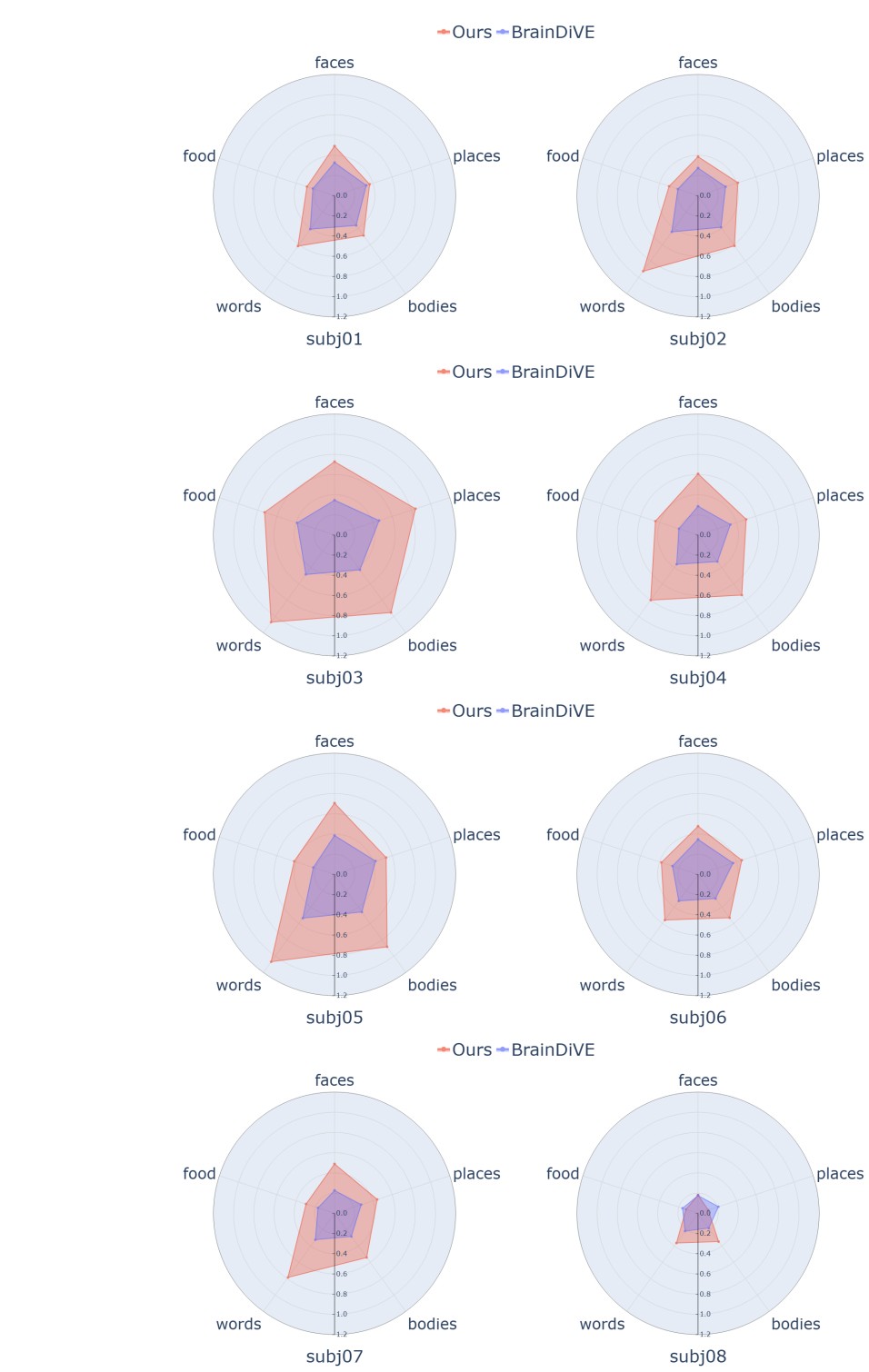

Figure 14: Neural activation alignment across semantic voxel groups.

## A.9 MORE RESULTS ON CROSS VALIDATION

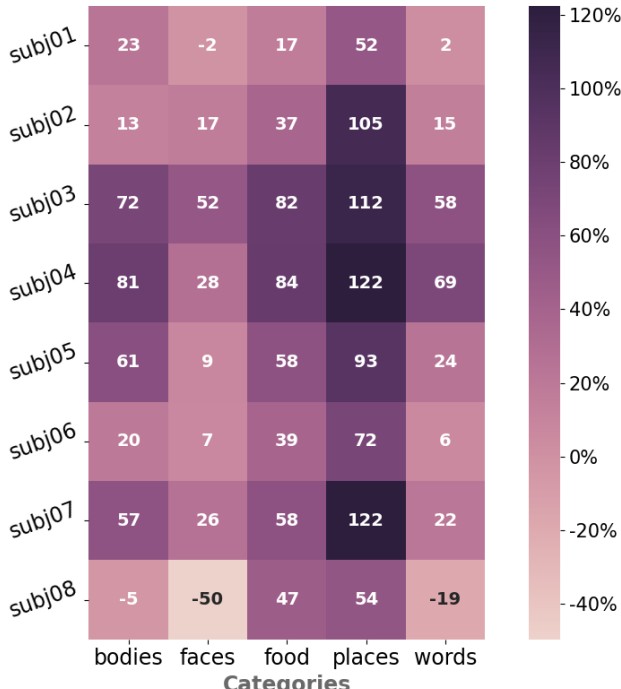

Figure 15: Delta heatmap shows the relative gain in predicted voxel activations (ours-baseline) / baseline when feeding the same generated images.

## A.10 PROMPTS FOR CLIP ZERO-SHOT CLASSIFICATION

This subsection details the text prompts, which are identical to those used in Luo et al. (2023a), for the zero-shot classification tasks reported in Table 1 and Appendix Table 2:

- bodies class = ["A photo of a torso", "A photo of torsos", "A photo of limbs", "A photo of bodies", "A photo of a person", "A photo of people"]

- faces class = ["A face facing the camera", "A photo of a face", "A photo of a human face", "A photo of faces", "A photo of a person's face", "A person looking at the camera", "People looking at the camera", "A portrait of a person", "A portrait photo"]

- food class = ["A photo of food"]

- places class = ["A photo of a bedroom", "A photo of an office", "A photo of a hallway", "A photo of a doorway", "A photo of interior design", "A photo of a building", "A photo of a house", "A photo of nature", "A photo of landscape", "A landscape photo", "A photo of trees", "A photo of grass"]

- words class = ["A photo of words", "A photo of glyphs", "A photo of a glyph", "A photo of text", "A photo of numbers", "A photo of a letter", "A photo of letters", "A photo of writing", "A photo of text on an object"]

Classification is performed by computing the cosine similarity between the CLIP latent of a generated image and the CLIP latents of the text prompts for each category. The image is assigned to the category whose prompt yields the highest similarity score.

## A.11 REPRODUCIBILITY STATEMENT

A complete public repository containing the full source code, trained model weights, and scripts to replicate all experiments will be released upon acceptance of this paper.

## A.12 USE OF LARGE LANGUAGE MODELS

This research utilized large language models (LLMs), specifically OpenAI's ChatGPT, in a limited and assistive capacity to improve efficiency in specific, non-substantive tasks. The models were used as tools under strict author supervision, and all final outputs remained under authorial control.

**Code Development and Debugging Assistance**  LLMs served as a coding assistant to accelerate development workflow. This involved:

- Clarifying syntax and usage examples for libraries such as PyTorch and Matplotlib.
- Proposing high-level debugging strategies for common errors.
- Generating good code structures, which were then extensively modified and integrated by the authors.

All code-related suggestions were treated as unverified proposals. They were rigorously reviewed, tested, and validated by the authors to ensure correctness and appropriateness before incorporation into the codebase.

**Visualization**  To develop the visualizations for this paper, we used ChatGPT for stylistic and organizational guidance. Furthermore, we asked it to generate an initial concept image, which served as the foundation for our final framework diagram.

**Manuscript Preparation and Polishing**  In the writing phase, LLMs were used exclusively for post-draft refinement. Their application was strictly limited to:

- Polishing the grammatical fluency, clarity, and conciseness of fully author-written sentences.
- Formatting tabular data from raw results into LaTeX code.

Crucially, no conceptual content, analysis, or original text was generated by the LLMs. All suggestions were critically evaluated and manually edited by the authors to ensure they accurately reflected the intended meaning and adhered to the academic tone of the paper.

