# OpenReview forum: "De-hallucinating CLIP Embeddings to Improve Brain-Vision Mapping"
_ICLR.cc/2026/Conference — Submitted to ICLR 2026_

### Official Review · Reviewer_bad3 · 2025-10-24

**Soundness:** 4
**Presentation:** 3
**Contribution:** 3
**Rating:** 6
**Confidence:** 5

**Summary:**

The paper proposes a lightweight learning pipeline to address hallucination problems of CLIP model (before voxel-wise regression to fMRI responses). Specifically, the proposed method first extracts object-centric CLIP features using segmentation masks and then stabilizes these diverse segment embeddings through attention modules with a concept bank of text-derived CLIP embeddings. The final step is fusing the concept-stabilized object features with the global embedding. Experiments on the NSD dataset show consistent improvements compared to standard CLIP regression.

**Strengths:**

* The manuscript identifies a concrete failure mode of CLIP-based brain encoding and decoding tasks, and uses segmentation, concept bank, and cross-attention to address it, which is straightforward, interpretable, and well-motivated.

* This work addresses an important and timely problem, i.e., bridging the semantic gap between foundation-model embeddings and human neural representations before brain encoding and decoding.

* While recent work [1] addressed a similar issue (semantic misalignment between supervision signals and neural recordings), the perspectives differ: [1] argues neural signals lacking full semantic information of supervision signals (CLIP embeddings), whereas this work treats this as the hallucination problem of CLIP. In both, the shared underlying goal is to close the semantic gap for better brain-model alignment.

  [1] Bridging the Gap between Brain and Machine in Interpreting Visual Semantics: Towards Self-adaptive Brain-to-Text Decoding, ICCV 2025.

**Weaknesses:**

* The pipeline relies on the image segmentation ground truth. In many natural datasets, segmentation may be unavailable or noisy. The paper does not analysis the sensitivity of the proposed method to segmentation errors and number of segments, such as using image segmentation methods instead of ground truth.

* While the paper emphasizes that CLIP’s semantic hallucination leads to misalignment with human brain responses, a closely related work (Mind-SA [1]) has already identified a similar semantic mismatch problem between brain and model representations. Despite the differing assumptions, both aim to bridge the same semantic gap between models and the brain before alignment. The paper does not explicitly discuss or compare with [1].

  [1] Bridging the Gap between Brain and Machine in Interpreting Visual Semantics: Towards Self-adaptive Brain-to-Text Decoding, ICCV 2025.

* The paper does not provide qualitative examples (as well as discussions) of image generation, and primarily compares its approach against a single baseline (BrainDiVE).

**Questions:**

How sensitive are results to prompt engineering or to the granularity of categories?

---

> ### Author Response · Authors · 2025-11-22
>
> We thank Reviewer bad3 for your constructive feedback. Below are our responses. (W stands for Weakness.)
>
> # [W1] Trade-off for Segments Quality and Availability
> We thank the reviewer for the insightful comment. We acknowledge that our method relies on object-level information and therefore has higher data-availability requirements compared to BrainDiVE. However, this additional training-free information is precisely what enables our model to achieve stronger semantic selectivity and improved performance. Grounding the encoding in explicit object and concept structure helps the model capture accurate object-centric scene content, mitigating CLIP's object hallucination tendency and producing a more biologically aligned embedding. Moreover, our model allows users to customize which objects, categories, or concepts to emphasize, enabling mappings tailored to specific neuroscientific or computational questions--a level of control not supported by BrainDiVE.
>
>
> When fine-grained annotations are unavailable, our method can flexibly use modern object-detection or segmentation models, such as SAM and YOLO. Even if these segments are imperfect (e.g., noisy, fragmented, or over-segmented), they typically do not substantially degrade performance. This is partly because CLIP often hallucinates or overgeneralizes object presence when visual evidence is weak, and ground-truth segmentations may not capture all objects that subjects attend to. Our framework mitigates these issues by extracting zoom-in features in addition to the global features, effectively increasing attention to semantically relevant regions even when boundaries are not precise. Furthermore, given that 3-second fMRI recordings in the NSD dataset are unlikely to preserve fine-grained spatial details or exact object boundaries, our approach is inherently robust to segmentation imperfections.
>
> In addition, our Concept Bank module, which can also support open-vocabulary, further stabilizes the mapping by reinforcing semantically relevant visual segments. Through cross-attention, object queries interact with a selected set of semantic concepts, which naturally suppresses segments that do not correspond to meaningful objects. In effect, the Concept Bank acts as a semantic filter: it amplifies concept-aligned regions while down-weighting noisy or spurious segments, providing robustness to both the quality and the number of extracted segments.
>
> To support the claims above, we additionally include comparisons that rely solely on object-detection methods to identify candidate objects. We also added a dedicated discussion on segmentation variability and robustness in the revised manuscript, as reflected in the updated PDF.
>
> # [W2] Comparison with Mind-SA
> We thank the reviewer for pointing us to Mind-SA (ICCV 2025). We became aware of this paper only after its release in October 2025, and it was therefore not available during the preparation of our ICLR submission (due September 2025). We have now included a detailed discussion of Mind-SA in our revised manuscript, summarized below.
>
> ## Fundamental Task Difference
> Our framework and Mind-SA are motivated and designed for different goals, operating under distinct problem formulations. The two pipelines address fundamentally different computational tasks with no shared evaluation protocol, making quantitative comparison an "apples-to-oranges" comparison:
> | | **Our Work** | **Mind-SA (ICCV 2025)** |
> | :--- | :--- | :--- |
> | **Task** | Image-to-Brain Regression | Brain-to-Text Decoding |
> | **Input** | Image | Image & fMRI |
> | **Output** | fMRI | Text |
> | **Goal** | To learn an improved visual-brain mapping that reveals semantic-level fMRI-vision relationships by enhancing CLIP features through de-hallucination. | To learn a subject-aware visual-brain mapping that leverages Gumbel-Softmax randomness to enable personalized text decoding. |
> | **Motivation** | CLIP image features often contain hallucinated objects that are not present in the image, which impairs alignment. | The brain does not allocate equal attention to all objects in an image, which harms decoding alignment. |
>
> ## Methodological Synergy
> Although the tasks differ, the two approaches are conceptually complementary. Mind-SA's subject-aware mechanism could enhance our object and concept selection, enabling personalized attention that further aligns the neural and visual modalities. Conversely, our de-hallucinated visual features could strengthen Mind-SA's visual-brain mapping, given that its CLIP visual encoder is also frozen. In principle, combining our improved visual features with Mind-SA's adaptive decoder could benefit both sides of the brain-vision pipeline. We have explicitly discussed this potential synergy in the final revision for our future work.

---

> > ### Author Response · Authors · 2025-11-22
> >
> > # [W3] Qualitative Examples & A Single Baseline
> > We appreciate the reviewer's observation. We would like to clarify that neither our method nor BrainDiVE is intended for reconstructing pixel-accurate images from neural activity (the traditional fMRI-to-image reconstruction task). Instead, both approaches aim to characterize the semantic level of neural responses by learning a brain-vision mapping, aiming to understand the neural functions. As one of the options to evaluate the quality of the mapping, we generate images to reflect the semantic tuning of specific voxel groups and focus on how well the generated images express the semantic content implied by the selected voxels, assessed through zero-shot classification rather than perceptual fidelity.
> >
> > ## Qualitative Samples
> > Because our model does not perform sample-wise fMRI decoding and is not trained to reconstruct real images, the generated samples are intentionally more abstract than those produced by conventional reconstruction methods (e.g., optimizing SSIM or pixel-level correlations). In response to the reviewer's interest, we have added qualitative examples to the supplementary materials in the revised submission.
> >
> > ## Baseline Selection
> > We primarily compared against BrainDiVE because it is the only existing method explicitly designed for semantic selectivity analysis via diffusion-based generation. Most other brain-vision models--such as MindEye, Takagi \& Nishimoto, and related reconstruction pipelines--focus on perceptual fidelity to specific stimuli and are not designed to evaluate the semantic tuning of voxel populations. Consequently, they are not directly comparable to our task. BrainDiVE remains the only established benchmark for semantic probing through generation, making it the appropriate baseline for our evaluation.
> >
> > To further validate our contribution beyond a single external baseline, we provided a rigorous ablation study (Section 3.6, Figure 7) and evaluated it as a retrieval task in Appendix A.2 and A.3, comparing our full model against:
> > - Image-only (Global CLIP): Represents the standard ``vanilla'' regression approach.
> > - Image + Object: Isolates the gain from segmentation, showing that our performance improvements stem specifically from our de-hallucination framework rather than extraneous factors.
> >
> > We also include a de-hallucinated variant of CLIP for comparison, further highlighting the advantages of our framework.

---

### Official Review · Reviewer_uttP · 2025-10-25

**Soundness:** 3
**Presentation:** 3
**Contribution:** 3
**Rating:** 4
**Confidence:** 4

**Summary:**

The authors present an interesting paper where they demonstrate that CLIP suffers from visual hallucinations (presence of concepts in images that are not actually there but either can be heavily associated in real world image statistics, but also might be quite erroneous and unexpected). The authors demonstrate a method they say effectively removes these visual hallucinations from the CLIP embeddings and then this provides a cleaner signal to link to fMRI data (specifically NSD) and that standard ridge regression results can be more informative as previous experiments that use non de-hallucinated CLIP embeddings might be making erroneous links via linking to information not actually presented in the image.

**Strengths:**

I really liked the goal of this paper. When I first read the abstract, it caught my attention and I felt that if the paper were to continue to fully investigate this, that this would be a paper I'd definitely be sharing with my colleagues and raising this point that I wasn't really aware of before now. I have previously examined this fMRI dataset using CLIP embeddings and it caused me to critically review how visual hallucinations might have affected some of these results. These papers do often get accepted into ICLR and this contribution I would deem to be highly relevant. Most notably, it would cause a lot of submissions to be critically analysed in light of these findings and that would be very relevant for progressing scientific knowledge on this field of research.

Having said that, the paper fell short in a few places and I really tried to find reasons to see past them, but some of them were quite important to me.

**Weaknesses:**

There are quite a few weaknesses I wish the authors had considered before submission, but I am hoping that by pointing them out, there will be a chance to make an updated version of the paper much better. My main problem is the evaluation of the method and the choice of experiments used to back up the claims of the paper. Let's take Figure 1, which very nicely shows the problem of visual hallucinations and how erroneous elements can creep into the predictions of constituent elements. I was fully expecting at this point to be seeing figures in the result section that compares and contrasts the old method with a de-hallucinated version of CLIP that could be specifically pointed at and where it is obvious that the new method removes the erroneous detection of these features. This would be an impactful easy win that is so intuitive, that actually its absence in the paper made me think this MUST HAVE been tried and didn't actually work. I can't contemplate why else such a comparison would be left out of the paper. This point needs to be clarified.

I also have a bit of an issue in understanding the logic behind the proposed multi-step de-hallucination method. I understand the extraction of the object-centric embeddings that are in the image and how you tied those to the concept bank of text embeddings, but why does simple fusion mean that the original problem no longer contains the hallucinated elements? You are expanding the representation and maybe adding more information from what you know is there, but that confounded information lurking in the background might still very well be there. This is where and exactly why I needed to see something visual to see that this method actually works. This is later described as a "corrective signal" but this logic needs to be explained more directly as I just don't see how it corrects incorrect hallucinations in any strong and direct way. At best it might upweight information related to the real present concepts, but the choice of language and claim is much stronger in what you have presented.

Your figure captions need a lot of work. Look at Figure 2. Consider the amount of information you have in there and look at that absolutely generic caption that tells me nothing new. I have circled the brain image in the top middle in my own notes and written, "What is this??" and I still have no idea what it represents. You have the details of the concept text encoder there, lots of letters and introducing variable names like "X^{s}" but I just don't know what these mean. I can make a guess but I shouldn't have to. I need to be able to be guided through a figure of such detail by relying on the caption.

Experimental details are often missing. Section 3.1. in the Dataset section, "all models are trained on a per-subject basis with a classic train/test split." - When I read this, I immediately went hunting for the specification of this because you absolutely cannot leave this information out in a paper like this. I have no idea what a classic train/test split because I've seen every manner of train/test splits and my experience tells me there is no such thing as a classic train/test split. Again, I shouldn't have to guess if this is 80/20 or something else because the issue of confounds across fMRI splits is also a huge issue in neural decoding.

Lines 356-357 highlight the core of the testing strategy and I just think this is pretty weak, forcing into only five categories and making a determination on something so highly intricate as to whether there is the presence of visual hallucinations. The test doesn't even look at the presence of this information, but rather increased scores on another task.

402-405: the category preferences are taken from Stigliani's localizers used in NSD, right? These are useful but hardly gold-standard mappings of where these categories are represented in the brain. I think this point should be considered and perhaps tone down the high claims of finding the "true" brain signal.

There is a lot of talk about image generation as a method of evaluation here, but I am pretty surprised to not see any visualised. It would be very helpful to see this, because otherwise I am just left to assume that the visualisation of these images is so poor that it was felt that not putting them in the paper was better than putting them in (and this raises some questions/doubts for me).

Also, there are some areas of language usage that need to be cleaned up a bit to make it more appropriate for scientific writing (e.g. "these evidences", "for 8 subject", "transparentizes", "the whoe model", "rather than cmodel omplexity").

**Questions:**

1. Can you clarify when talking about the global embedding, what exactly you mean. Are you taking the CLS token of the final layer of the vision encoder or doing some merging of all the patch tokens. This is important to know and it's not specified in the paper.

2. Can you explain why you didn't visually contrast hallucinating CLIP with the de-hallucinated version showing that the de-hallucinated version has a more sensible ranking order of elements that are detected (an adaption on the problem that Figure 1 demonstrates)?

3. Can you provide a more extensive / simpler / extended (whatever) explanation of how adding more information to CLIP's global embedding effectively "de-hallucinates" the imagined concepts? I am really not clear how this works. I really hope it does, but the lack of a direct visual contrast (see (1)) makes me quite suspicious. I want this to be the case, but I need to be convinced first.

4. I presume you used ridge regression in the brain-vision mapping but I actually see no mention of this in the equations. I don't know a single other work that has had success in brain-vision mapping without providing some sort of extra constraint in the regression. I can't believe you didn't use one as it's so non-standard, but it's not mentioned in the equation. Can you please clarify this point? If you didn't use ridge regression, I'm going to need an extremely strong and well-defended motivation as to why this choice was made.

5. I just wanted a direct answer to the question of whether this paper is the one introducing the idea of visual hallucinations in CLIP and this isn't mentioned in other work that you are building on? The reason is that this idea in itself is quite big and I thought probably needs way more investigation to support the claims and it's an incredibly relevant result to the community. I'm very eager to see this finding spread out in the community, but it almost seemed like a secondary finding in this submission. That is fine, if it's not your choice to want to lead with that claim, but I needed to be sure of this myself during my review. [EDIT] : I see now this is explained in A.1. but this is not linked directly in the main paper, causing my earlier confusion. This absolutely needs to be amended because it's almost deliberately unclear in the current formulation.

6. On Line 165, you say the entire scene can be decomposed into into the labelled segments, but is that truly the case the entirety of each image is fully segmented such that every pixel belongs to a single segment? I thought backgrounds and complex patterns would not be assigned to a specific categorical segment. If my understanding is correct, then it's not the case that: "[e]ach image can be decomposed as X = ....." (rest of line 165). Please can you clarify this point?

7. Do you have any statistical tests to report for section 3.2? Do we just accept the trends in Figure 4 on varying 1-4% scales are meaningful? Looking at Figure 3 and having those high concentrations around zero with (also) no statistics makes me just wonder about the robustness of the results.

8. The end of section 3.2. contains a claim that summarises that go some way to explain why the reported differences occur, but I want a more extensive rationale. Can you be specific, pointing specifically to the presence of hallucinated objects in CLIP representations, that would make scores worse on this test? I'm interested to understand if you can tie it to the additional corrected embedding and how this works.

9. Can we see any of the generated images that were fed into the encoder to compare your method to BrainDiVE?

10. In Line 411-412, you say subject 6 had fewer training samples relative to other subjects. I went back to review your Dataset description and you say that each subject saw 10,000 images and there is no description anywhere else in the paper that leads to any information on why some subjects might have fewer data samples than others. What's going on here?

---

> ### Author Response · Authors · 2025-11-24
>
> We thank Reviewer uttP for your constructive feedback. Below are our responses. (W and Q stand for Weakness and Question, respectively.)
>
> # [W1 + Q2] Visual Comparison of Original vs. De-hallucinated CLIP
> Thank you for raising this important point. We agree that **sample-wise** visual comparisons between the original CLIP ranking and our de-hallucinated version would be **highly intuitive**. In response to your suggestion, we have added several representative visual comparisons in the revised `Appendix A.6`. These examples complement the quantitative analysis and make the effect clearer on individual images.
>
> In addition, we want to clarify why we did not originally include only visual comparisons. **Showing only a handful of visual examples might be interpreted as cherry-picking**, and would not convincingly demonstrate that our de-hallucination strategy generalizes across the dataset. For this reason, instead of relying solely on isolated visual cases, we decided to provide **a quantitative dataset-level evaluation** by reformulating the `Figure 1` ranking scenario as a **retrieval task**.
>
> ## What the retrieval task does in simple terms
> For every image in the NSD test set:
> - We take the ground-truth list of objects appearing in the image.
> - We compute the similarity between the visual embedding and text prompt embeddings.
> - We examine whether these true objects appear near the top of the ranked similarity list.
>
> This procedure is **identical** to building `Figure 1`, but applied **systematically across the full evaluation set** rather than on individual examples.
>
> ## Why this demonstrates de-hallucination
> If CLIP hallucinates **objects not present** in the image, these **incorrect objects** will receive **high similarity scores** and appear near the **top** of the retrieval ranking. If our method suppresses hallucinated semantics, then:
> - True objects should move **up** in the rankings.
> - Spurious/hallucinated objects should move **down** or disappear.
>
> Thus, retrieval-quality metrics provide a **principled** way to measure whether the representation becomes **less hallucination-prone** on a large scale.
>
> ## What we observe
> As shown in `Appendix A.3` (and summarized below), our de-hallucinated embedding consistently retrieves **more correct objects** and **fewer spurious ones** across **thousands* of images. The improvements hold under all four retrieval metrics we evaluate, demonstrating both **robustness** and **effectiveness** of the de-hallucination mechanism.
> |  | Recall | mAP | AUC | nDCG |
> |--------|--------|-----|-----|------|
> | CLIP   | .733   | .695 | .830 | .780 |
> | Ours   | .822   | .765 | .836 | .784 |
>
> # [W2 + Q3 + Q8] Expanded Explanation and Rationale for Our Method and Results
> Thank you for raising this point. We agree that a clearer rationale is needed. In the original CLIP global embedding (CLS token), hallucinated objects often receive **artificially high representation strength**, even when they do not appear in the image. These spurious semantic dimensions behave as noise in the encoding model because they introduce features that have **no corresponding neural activation** in the fMRI signal. This forces the regression to fit variance that the brain never encoded, effectively lowering the **signal-to-noise ratio** of the mapping and reducing voxel-prediction performance.
>
> We are not claiming that our fused representation fully eliminates hallucinations. Instead, our strategy mitigates this issue by incorporating **object-level information extracted from ground-truth segments**. This explicitly **boosts the representation of objects that are truly present** in the image--the same objects the brain reliably responds to--and reduces the influence of semantic content that is unsupported by the visual scene. Because hallucinated objects do not correspond to any segmented region, their contribution to the final representation is naturally **suppressed** by the object-driven refinement. The resulting embedding contains **a cleaner and more accurate semantic profile**, where true-object features are strengthened and hallucinated components are attenuated. This improves alignment with the brain's object-selective responses and therefore explains the performance gains observed in `Section 3.2` and `Section 3.3`.
>
>
> # [W7 + Q9] Visual Examples of Generated Images
> We have added qualitative images generated from our method and BrainDiVE in `Section 3.3` and `Figure 5` in the revised PDF. These visualizations help illustrate the improvement in semantic selectivity.

---

> ### Author Response · Authors · 2025-11-24
>
> # [W3 + W6 + W8] Adjustment for Figure Caption, Tone, and Typos
>
> ## Caption for Figure 1
> Thank you for pointing this out. We agree that the caption for Figure 2 was not sufficiently informative given the complexity of the diagram. In the revised version, we have updated the figure with a fully self-contained caption that explains all variables, modules, and notations appearing in the illustration. We appreciate this suggestion as it meaningfully improves the clarity and accessibility of the paper.
>
> ## Tone adjustment for Localizers
> Thank you for pointing this out. You are absolutely right that the category-selective ROIs derived from Stigliani et al. localizers--while widely used in NSD and prior encoding work--are not gold-standard or exhaustive maps of categorical representations in the human brain. They provide a functional approximation that is useful for benchmarking, but they should not be interpreted as capturing the full or definitive set of category-selective signals.
>
> Our intention was not to claim that we have discovered the "true" brain signal, but rather that our refined representation aligns more closely with the known category-selective patterns measured by these established localizers. We agree that this nuance was not made sufficiently clear in the original text.
>
> In the revision, we have:
> - Provided explicitly acknowledge that the Stigliani localizers represent a useful but incomplete model of category-selective organization.
> - Toned down phrasing that suggests these regions reflect the ground-truth neural representation.
> - Emphasized that our findings demonstrate improved consistency with known functional gradients, not the ground-truth neural tuning.
>
> We appreciate the reviewer bringing attention to this point, as it helps us clarify the scope and interpretation of our results.
>
> ## Typos
>
> We appreciate your careful reading. We have gone through the manuscript again and corrected all typographical and grammatical issues we identified, including the examples you highlighted. We have tried our best to ensure that the revised version uses precise and appropriate scientific language throughout.
>
>
> # [Q1 + Q4 + Q6] Clarification on Technical Details
> Thank you for raising this clarification. We have clarified in our updated version.
>
> ## Global Embedding
> In our paper, ""global embedding""' refers specifically to the CLS token from the final layer of CLIP's ViT image encoder in the shared space, without any additional pooling over patch tokens. We have explicitly stated this in Section 2.1 to avoid ambiguity.
>
> ## Ridge Regression
> Thank you for pointing this out. We indeed used L2-regularized linear regression for the brain-vision mapping. In our implementation, this is expressed as linear regression with a weight-decay coefficient of 0.8, which is mathematically equivalent to ridge regression. We apologize that the explicit regularization term was not shown in the simplified equation in Section 2.4. We will revise the equation and text to clearly state that our encoding model uses ridge-style L2 regularization, consistent with standard practice in prior NSD and brain-encoding studies.
>
> ## Image Decomposition and Segment Coverage
> Thank you for this great observation. You are absolutely correct that the instance annotations in MS-COCO do not necessarily cover the entire image--backgrounds, textures, or complex patterns often remain unlabeled. We explicitly considered this when designing our model. In addition to the provided ground-truth instance masks, we did introduce an extra background segment that includes all remaining pixels. This ensures that the segmented representation covers the entire image and prevents loss of contextual information. This background segment is treated like any other segment during cross-attention, allowing the encoder to incorporate global scene context while still preserving the object-level structure from the annotated instances. We have revised Line 165 to make this construction clear.
>
> # [Q7] Statistical Significance
>
> Thank you for raising this important point. Each data point in Figure 4 represents the average $R^2$ improvement computed over 1,000 samples, not a single measurement, which already provides some robustness to variability. Nevertheless, we fully agree that statistical testing should accompany these trends to establish their significance more rigorously.
>
> In the revised version, we have included a paired t-test (and report corresponding p-values) comparing baseline CLIP embeddings versus our de-hallucinated embeddings across samples. These tests consistently indicate that the observed improvements are statistically significant across subjects. We have updated these results in Section 3.2 to better highlight the reliability of the findings.

---

> ### Author Response · Authors · 2025-11-24
>
> # [W4 Q10] Clarification on Train/Test Split and Subject-Specific Sample Counts
> Thank you for highlighting these issues. We apologize for the lack of clarity in the original submission.
>
> ## NSD Classic Train/Test Split
> In our paper, ``classic train/test split'' refers to a convention that has become standard in NSD-based studies across fMRI-to-image reconstruction, caption decoding, and semantic selectivity analysis:
> - 1,000 images are commonly viewed by all eight subjects.
> - These 1,000 shared images form the test set.
> - The remaining 9,000 images (unique to each subject) form the training set.
>
> This split is widely used in prior works, which is why we referred to it as the ``classic'' NSD split. We have revised the Dataset section to state the exact numbers explicitly so that the evaluation protocol is clear without requiring prior knowledge of NSD practices.
>
> ## Subject-Specific Sample Counts
> In NSD, although each subject is intended to complete 40 sessions (10,000 images), the actual number of usable sessions varies across subjects due to session interruptions, motion-related exclusions, and other data-quality issues. This variability is documented in both the NSD paper and the accompanying technical notes. Specifically, we have publicly available data:
> - 40 sessions (full 10,000 images): subj01, subj02, subj05, subj07
> - 32 sessions: subj03, subj06
> - 30 sessions: subj04, subj08
>
> Thus, subj06 indeed has fewer training samples (32 sessions instead of 40), which explains the reduced number of valid trials compared to subjects with complete datasets. We have updated the Dataset section to clearly explain this variability and prevent confusion.
>
> Thanks again for pointing these out.
>
> # [W5] Evaluation Choice
> Thank you for raising this concern. We agree that visual hallucination is a complex phenomenon and cannot be fully characterized by reducing the evaluation to only five broad categories. However, we would like to clarify the scope and intention of our testing strategy.
>
> Our main focus is not to propose a new general-purpose de-hallucination method for CLIP. Instead, our goal is to improve brain-vision mapping by mitigating the impact of hallucinated semantics in CLIP's representation. For this reason, we follow BrainDiVE and use image generation and downstream semantic selectivity evaluation as the primary way to assess the quality of the mapping. These tests measure how well the refined representation aligns with voxel tuning and therefore target our central objective.
>
> Because we are not aiming to develop or benchmark a full de-hallucination framework, we did not include an extensive hallucination-specific evaluation (e.g., hallucination detection accuracy). That said, we acknowledge the value of analyzing the hallucination effect more directly. To provide additional clarity, we have included in the appendix a retrieval-style analysis over the entire dataset, which quantitatively evaluates whether the refined embedding prioritizes true objects over hallucinated ones. This formulation is a more systematic and scalable alternative to the limited five-category analysis and directly reflects the semantic correction we aim to achieve.
>
> We have revised the paper to clarify this evaluation design and better distinguish our focus on improving neural encoding rather than benchmarking de-hallucination as an independent task.
>
> # [Q5] Visual Hallucination in CLIP
>
> Thank you for raising this important point. Hallucination is indeed a well-known issue in large language models and in cross-modal models such as CLIP, and we are not the first to observe hallucination in pre-trained CLIP variants. However, to the best of our knowledge, we are the **first** to investigate how these object-level hallucinations specifically affect brain-vision alignment--a setting in which CLIP is currently the dominant image encoder used for neural encoding, decoding, and reconstruction research.

---

### Official Review · Reviewer_rgXV · 2025-10-27

**Soundness:** 2
**Presentation:** 3
**Contribution:** 3
**Rating:** 4
**Confidence:** 5

**Summary:**

This paper focuses on the "hallucination issue" of CLIP visual embeddings in brain-vision mapping tasks (i.e., encoding objects not present in the image). It proposes a lightweight de-hallucination framework consisting of "object-level representation extraction + concept bank anchoring + global feature fusion" and validates, on the NSD fMRI dataset, that this framework enhances neural encoding accuracy and category-selective voxel activation.

The work integrates the de-hallucination issue of vision-language models with the need for brain representation alignment in neuroscience, making its research topic both innovative and of interdisciplinary value. The methodological design aligns with biological intuition (integrating bottom-up perception and top-down categorical knowledge), and the core conclusions are supported by experiments. However, there remains significant room for improvement in aspects such as the completeness of qualitative validation, the comprehensiveness of quantitative comparison, and the rigor of methodological details. Critical supplementary experiments are required to enhance the persuasiveness and generalizability of the conclusions.

**Strengths:**

1. The authors have identified the phenomenon of inconsistency between conceptual representations and image content in CLIP. Building on this issue, they propose a method that does not incur substantial computational overhead to mitigate such hallucinations.

2. Experimental results on neural coding demonstrate the effectiveness of eliminating hallucinations in CLIP for improving coding accuracy.

**Weaknesses:**

1. No supplementary comparative diagrams of hallucinations between "the proposed method vs. original CLIP" (e.g., improved Top-8 prediction results corresponding to Figure 1) are provided, making it impossible to intuitively judge changes in the ranking of correct objects and the reduction of spurious/contextual hallucinations.

2. For the generated results in Table 1, only numerical data (zero-shot accuracy) is presented, without visual comparisons of generated images between "the proposed method vs. BrainDiVE," making it difficult to reflect the practical significance of improved semantic consistency.

3. No dimensionality-reduced visualizations (e.g., t-SNE or UMAP) of the original CLIP global embeddings, object-level embeddings ($Z_{obj}$), and concept-enriched embeddings ($Z_{obj}^C$) are provided. The de-hallucination mechanism of the method becomes a "black box," resulting in insufficient interpretability.

4. No comparisons are made with classic methods (e.g., CLIP-Guided Decoding [1], CLIP-DPO [2] ) on general CLIP de-hallucination benchmarks (such as POPE [3] , object-level comparison tasks in Liu et al. (2024) [4] ), making it impossible to prove the generality of the de-hallucination capability.

5. Incomplete ablation experiments: Only the comparison of "image→image+object→full" is conducted, without ablating the concept bank, cross-attention, or object segmentation, nor verifying different fusion strategies. This makes it impossible to distinguish the contributions of individual modules.

6. The selection of the concept bank prompt ("a cropped centered photo of [object]") lacks justification: No comparison of the effects of other prompts is provided, nor is the generalization to "unseen categories" verified, which may affect reproducibility.

7. The simplicity of the fusion strategy ($z=(z_{img} + z_{obj})/2$) lacks support: No comparisons with alternative fusion methods (e.g., attention weighting, adaptive weighting) are made, nor is a sensitivity analysis of fusion weights conducted, making it impossible to prove its optimality.

[1] Seeing is Believing: Mitigating Hallucination in Large Vision-Language Models via CLIP-Guided Decoding

[2] CLIP-DPO: Vision-Language Models as a Source of Preference for Fixing Hallucinations in LVLMs

[3] Evaluating Object Hallucination in Large Vision-Language Models

[4] Investigating and Mitigating Object Hallucinations in Pretrained Vision-Language (CLIP) Models

**Questions:**

Overall, this paper is good in both topic selection and quality. If the authors can address and supplement the issues I raised in the "Weaknesses" section, I will consider raising the score to 6.

---

> ### Author Response · Authors · 2025-11-24
>
> We sincerely appreciate Reviewer rgXV for your valuable feedback. Below are our responses. (W stands for Weakness.)
>
> # [W1] Comparison with CLIP
>
> Thank you for this helpful suggestion. We would like to clarify that comparing our method vs. original CLIP is effectively the **same** as comparing our method vs. BrainDiVE, because BrainDiVE uses the **unaltered** CLIP global image embedding as its input for the regression model. Thus, the difference between our method and BrainDiVE directly reflects the difference between our method and the original CLIP embedding. This aligns with our core motivation to de-hallucinate CLIP representations and thereby improve visual-brain mapping performance.
>
> In the main paper, our ablation results in `Section 3.6` already report the zero-shot classification improvements of our enhanced embedding compared to vanilla CLIP. Furthermore, in ``Appendix A.3``, we extend the illustrative example shown in Figure 1 to the complete NSD dataset by formulating it as a **retrieval task**, evaluating all three levels of representation (CLIP global, object-level, and concept-enriched) using standard ranking-based retrieval metrics. These analyses provide a quantitative assessment of the reduction in hallucinated features.
>
> We agree that showing **a direct visual example** analogous to Figure 1 would make the effect more intuitive. Therefore, we have added representative side-by-side examples of CLIP vs. our de-hallucinated embedding in ``Appendix A.6`` to make our contribution more visually intuitive.
>
> # [W2] Qualitative Examples
>
> We have added qualitative images generated from our embedding and BrainDiVE in the main paper in ``Figure 5``. These visualizations, along with a paragraph of analysis in ``Section 3.3``, help illustrate improvement in semantic selectivity.
>
> # [W3] Interpretability and Visualization
>
> Thank you for raising this constructive suggestion. We agree that dimensionality-reduced visualizations can help illustrate the differences between CLIP's original global embeddings and our de-hallucinated embeddings. In ``Appendix A.5``, we now include a UMAP visualization showing that the refined embedding moves toward the true concept. This helps clarify the de-hallucination process and reduces the black-box impression.
>
>
> # [W4] Comparison with General CLIP De-hallucination methods
>
> Thank you for this thoughtful suggestion. Our work focuses specifically on how CLIP object-level hallucinations adversely affect the construction of brain-vision encoding models, where CLIP's image embedding is a central component in current neural modeling pipelines. Our goal is not to develop a general-purpose CLIP de-hallucination method for image-text tasks, so generalization to multimodal benchmarks is not the primary focus of our paper.
>
> That said, we agree that including a comparison to an existing de-hallucination strategy helps clarify the distinct contribution of our object-concept refinement. In the revised appendix, we include a comparison using the retrieval-style representational similarity analysis (the same formulation as in Appendix A.3). This allows us to evaluate our embedding against a representative de-hallucination baseline under the same similarity framework we use throughout the paper. The results confirm that our method corrects hallucinated concepts more effectively in the context of neural encoding.
>
> # [W5-7] Incomplete Ablations
>
> Thank you for highlighting the need for more fine-grained ablations. We agree that disentangling the contributions of each component would strengthen the paper. In the revised version, we extend our ablation study with the following additional analyses in ``Appendix A.4`` and summarized below:
> - **Concept Bank Prompt Ablation**: To test the robustness of the concept bank design, we included three more prompts **("a photo of [object], "a detail view of [object]", "an isolated [object]")**. This contrasts with our original **"a cropped centered photo of [object]"** and allows us to evaluate how sensitive the concept representations are to prompt style.
> - **Fusion Strategy Ablation**:
>     - 0.25 (global) + 0.75 (object-concept)
>     - 0.75 (global) + 0.25 (object-concept)
>
>   This analysis helps determine whether the chosen weights are optimal or whether alternative balances offer advantages.
> - **Cross-Attention Ablation**: Different numbers of attention **heads (2, 4, 6, 8)**
>
>   These experiments clarify whether performance gains stem from the cross-attention mechanism itself or from the learnable projection layers.

---

> > ### Comment · Reviewer_rgXV · 2025-11-27
> >
> > Thank you for your supplements. However, regarding the "Qualitative Examples," I have not found the generated image results in the appendix of the manuscript, nor have I located any downloadable supplementary materials under this page. Could you please inform me where I can access this section?

---

> > > ### Author Response · Authors · 2025-12-02
> > >
> > > Thank you for your continual interest in our work. The **Qualitative Examples** can now be found in `Section 3.3` and `Figure 5` in the revised pdf.

---

### Official Review · Reviewer_TtCt · 2025-10-28

**Soundness:** 2
**Presentation:** 3
**Contribution:** 2
**Rating:** 2
**Confidence:** 4

**Summary:**

This paper introduces a novel de-hallucination framework designed to mitigate the semantic hallucinations present in CLIP’s global visual embeddings when applied to brain-vision mapping tasks. The method first extracts object-level embeddings from segmentation masks, then aligns them with a text-derived “concept bank” using cross-attention, and finally fuses them with global scene embeddings to form de-hallucinated visual features. Experiments on the Natural Scenes Dataset (NSD) show improved voxel-wise regression, stronger activation in semantically selective regions, and better semantic consistency in brain-driven image generation compared to BrainDiVE.

**Strengths:**

1. Original Problem Definition
The paper clearly identifies and analyzes a real issue—hallucinatory semantics in CLIP embeddings—within brain-vision applications. This is an underexplored but critical limitation for neuroscience-aligned visual models.

2. Strong Empirical Validation Results on NSD are extensive and multi-faceted: voxel-wise R² improvement, category-based image generation, and neural activation alignment. Demonstrates neuroscientific relevance.

**Weaknesses:**

1. Missing Formalization of the “De-Hallucination” Objective：

The authors introduce cross-attention between object features and a “concept bank” but do not specify what loss function or alignment constraint guides this process—e.g., whether it minimizes distance between object embeddings and concept embeddings, or uses contrastive or reconstruction objectives.

Without a defined objective, it’s unclear how the model learns to “de-hallucinate” beyond architectural heuristics. Adding even a small formal definition (e.g., minimizing semantic divergence) would make the contribution more rigorous.

2. Relation to Variational and Contrastive Frameworks Is Unclear：

The cross-attention formulation (Eq.6) suggests an implicit distribution alignment between noisy visual segments and stable semantic priors—conceptually similar to variational information bottleneck or contrastive predictive coding.
However, the paper does not clarify whether the model explicitly minimizes KL divergence, employs negative sampling, or estimates mutual information.

Clarifying this connection would strengthen the theoretical grounding of the “concept stabilization” process.

3. Quantitative Gains Are Modest and Require Statistical Analysis:

While the paper reports modest improvements (max average R² increase ≈ 1.5%), there is no mention of statistical testing (e.g., permutation test or paired t-test).

Given the small margins, readers cannot assess whether the observed trends are robust across subjects.
It would help to include confidence intervals or significance tests for voxel-wise gains.

4. Lack of Comparison Against Stronger Baselines

Only BrainDiVE is used for comparison. BrainDiVE (Luo et al., 2023) is an appropriate baseline, but recent NSD-based methods such as MindEye2 (Scotti et al., 2024b), CLIP-MUSED (Zhou et al., 2024), and NeuroCLIPs (Gong et al., 2024) should be included or at least discussed.

This would clarify whether the proposed improvements stem from de-hallucination or from differences in architecture or training scale.

5. Ambiguity in Image Generation Evaluation:

While CLIP-based evaluation is convenient, using the same model (CLIP) both for de-hallucination and evaluation may confound results.
Alternative metrics—such as human-annotated category accuracy or FID computed against category exemplars—would provide more independent validation.

6. Missing Details on Cross-Attention Implementation:

Readers need to know whether it is a single-head or multi-head attention, whether weights are learned or frozen, and how it is regularized.
Given that this is the core mechanism of the framework, a concise architectural schematic or pseudocode would enhance clarity.

7. Need for Perceptual or Behavioral Validation:

The argument that the proposed method produces representations “aligned with human perception” is well-motivated but not empirically validated beyond neural alignment.

Even a small-scale human annotation (e.g., asking whether generated images reflect the correct object categories) would substantiate the claim of perceptual fidelity.

8. Overlap with Recent Works:

While the paper distinguishes itself by focusing on hallucination mitigation, several related works also incorporate semantic grounding or object-based refinement.

A clearer discussion of differences (e.g., whether the concept-bank cross-attention introduces unique mechanisms) would help position this paper more distinctly.

9. Ablation Study Interpretation

Although adding concept-level features improves mean performance (54.0%), for some subjects (e.g., subj08), performance drops.
This should be analyzed—perhaps due to segmentation quality or concept-bank mismatch. A per-subject discussion would strengthen the robustness claim.

**Questions:**

See Weakness.

**Details Of Ethics Concerns:**

N/A.

---

> ### Author Response · Authors · 2025-11-24
>
> We thank Reviewer TtCt for your constructive feedback. Below are our responses. (W stands for Weakness)
>
> # [W1 + W2] Clarification on goal and method
>
> Thank you for raising this point. We believe there may be a misunderstanding regarding the goal of our work. Our aim is not to solve CLIP de-hallucination by fine-tuning CLIP with new contrastive, variational, or KL-based objectives to update CLIP's weights. Instead, our focus is on learning a visual-brain mapping to understand the semantic-level relationships between fMRI voxels and visual concepts. Rather than directly using CLIP's visual features, we design a fine-tuning-free de-hallucination module that works jointly with the visual-brain mapping function. Our motivation is to inject potential local concept features into CLIP's global representations to improve the visual-brain mapping.
>
> More importantly, the two-stage design allows us to analyze brain activation at the semantic level, which is the central focus of our work. Our de-hallucination approach for CLIP is training-free: we explicitly inject ground-truth local object features (or SAM-derived segments when ground truth is unavailable) by mimicking a zoom-in mechanism to guide attention. We also introduce a concept bank--optionally open-vocabulary--that further refines features through cross-attention. Overall, only the cross-attention weights and the visual-to-brain mapping are trained, keeping the model lightweight and parameter-efficient. Its behavior is driven solely by the voxel-prediction objective, without relying on contrastive, reconstruction, or variational losses.
>
>
> We will clarify this distinction in the revision to avoid the impression that our method is intended as a new training-based de-hallucination framework.
>
> # [W4 +W8] Comparison Against Stronger Baselines and Recent works
>
> We would like to clarify that neither our method nor BrainDiVE is designed to reconstruct pixel-accurate images from neural activity, as in traditional fMRI-to-image reconstruction tasks. Instead, both approaches aim to characterize the semantic level of neural responses by learning a brain-vision mapping to better understand neural function. To assess whether our mapping captures meaningful semantic correspondence between visual and fMRI signals, we modulate specific voxel groups to generate images. The zero-shot classification step is used only to verify whether these generated images contain the expected semantic content--not to evaluate pixel-level fidelity.
>
> BrainDiVE is currently the only method explicitly designed for semantic selectivity analysis via diffusion-based generation. In contrast, MindEye2 focuses on fMRI-to-image reconstruction, CLIP-MUSED is tailored for fMRI-based multilabel classification, and NeuroCLIP targets fMRI-to-video reconstruction. These approaches are not designed to probe semantic tuning at the voxel population level and are therefore not directly comparable to our task. Please refer to table  below. BrainDiVE remains the only established benchmark for semantic probing through generation, making it the appropriate baseline for our evaluation.
> | Method       | Input | Output | Task                          |
> |--------------|-------|--------|-------------------------------|
> | MindEye2     | fMRI  | Images | fMRI-to-image reconstruction  |
> | NeuroCLIP    | fMRI  | Videos | fMRI-to-video reconstruction  |
> | CLIP-MUSED   | fMRI  | Predicted Labels | fMRI-based multilabel classification |
> | **BrainDiVE**| Images| fMRI   | neural semantic selectivity   |
> | **Ours**     | Images| fMRI   | neural semantic selectivity   |

---

> ### Author Response · Authors · 2025-11-24
>
> # [W5 + W7] More Evaluation Metrics for generated images
>
> Thank you for raising this important point. We agree that independent evaluation metrics are valuable, but both human annotation and FID-style measures present practical or conceptual limitations in our setting.
>
> ## Why FID is not applicable in our task
> FID requires comparing generated images to a real image distribution. Our method, however, is not performing pixel-level reconstruction. The generated samples are intentionally abstract. They are meant to visualize the semantic selectivity of voxel groups, not to reproduce real images or match any natural image distribution. For this reason, there is no real-image ground truth for these semantic generations. FID would measure pixel-level alignment that is irrelevant to our objectives, and it would penalize the model for not matching distributions that we do not aim to reproduce. This is consistent with prior semantic-only generation works (e.g., BrainDiVE), which also avoid FID for the same reason.
>
> ## Why human annotation is difficult at this scale
> The number of generated samples is large (5000 for each subject), and manually labeling them consistently requires substantial time and resources. In addition, collecting human evaluations often triggers IRB and ethical review requirements, especially for brain-encoding studies, which cannot be completed within the rebuttal timeline.
>
> Given these constraints, CLIP-based semantic evaluation remains the most practical and widely adopted measure for capturing semantic correctness rather than pixel fidelity.
>
> ## Complementary analyses
> We agree with the idea of the suggestion. To strengthen the evaluation, we include a CLIP-based retrieval analysis (Appendix A.3) and provide qualitative examples that directly show how our method suppresses hallucinated concepts. These additions offer more evaluations while remaining aligned with the semantic nature of our task.
>
> # [W6] Cross Attention Details
> Thank you for pointing this out. We agree that the implementation details of the cross-attention module were not sufficiently described, and we appreciate the suggestion.
>
> In the revised paper, we have added a concise description of the module in the Implementation Details section. To clarify here:
> - Attention type: We use a single-head cross-attention module.
> - Learnable parameters: All K/Q/V projection matrices are trainable, while CLIP's vision and text encoders remain frozen.
> - Regularization: The cross-attention output is regularized only through the ridge regression objective used in the downstream voxel-prediction model; no additional losses or constraints are applied.
>
> # [W9] Interpreting Subject-Wise Variation in Ablation Results
>
> Thank you for this observation. We agree that subject-level differences needs discussion, especially in the context of our ablation results.
>
> A key factor is that NSD subjects differ substantially in data quality and the number of usable sessions, as documented in both the NSD paper and technical notes. In particular, subj08 is known to have the lowest fMRI signal quality and completed only 30 usable sessions, compared to 40 for the highest-quality subjects (subj01, subj02, subj05, subj07). This reduced data quantity and lower SNR make subject-specific mappings inherently less stable, and performance fluctuations in downstream tasks are expected.
>
> A second factor is differences in the distribution of visual stimuli and object occurrences across subjects’ unique image sets. Because our method explicitly leverages object-level and concept-level semantics, mismatches in object frequency and category coverage can influence the quality of the learned mapping. Subjects with fewer sessions or skewed object distributions benefit less from concept-level refinement, and this effect is most pronounced for subj08.
>
> Overall, these subject-wise differences reflect properties of the underlying dataset rather than inconsistencies in the method itself. We will expand the per-subject discussion in the revision to make these sources of variability clear.

---

### Meta-Review · Area_Chair_2frc · 2025-12-15

**Summary:**

This paper proposes a framework to mitigate the hallucination of semantics in CLIP's global visual embeddings for improving brain-vision mapping. The method integrates object-level features from segmentation masks with a text-derived concept bank using cross-attention, then fuses them with the global CLIP embedding to form a de-hallucinated representation for voxel-wise regression. Experiments on the NSD dataset report improved alignment with category-selective brain regions. Reviewers provided mixed scores: one strong positive (6), two marginally below acceptance (4), and one reject (2). The primary strengths identified include a novel, well-motivated problem at the intersection of neuroscience and foundation models, and a straightforward, interpretable method. Key concerns centered on insufficient empirical validation of the core de-hallucination claim, missing comparisons with recent and relevant baselines, lack of statistical analysis for modest performance gains, and insufficient implementation details and ablation studies. In the rebuttal, the authors clarified that their goal is not general CLIP de-hallucination but improving brain-vision mapping specifically, and they are not aiming for pixel-level image reconstruction. They added new qualitative examples comparing their method with BrainDiVE and CLIP, provided more implementation details, and included additional ablation studies on prompts and fusion strategies. They also argued that other recent methods (e.g., MindEye2) are not directly comparable as they focus on different tasks. However, fundamental issues regarding the direct demonstration and quantitative validation of the de-hallucination effect, as well as the robustness and significance of the improvements, remain largely unaddressed.

**Reviewer Concerns:**

The rebuttal partially addressed concerns about missing qualitative examples and some implementation details. However, major outstanding concerns persist, particularly the lack of direct, convincing evidence that the method effectively "de-hallucinates" CLIP embeddings and that this directly leads to the reported neural alignment improvements. The need for statistical validation of modest gains and a clearer discussion against relevant recent works also remains.

**Reviewer Scores:**

Given the rebuttal's focus on clarifications rather than substantial new evidence to alleviate core methodological and validation concerns, it is unlikely the reviewers with scores of 4 (rgXV, uttP) would raise their scores significantly. The reviewer with a score of 2 (TtCt) would likely maintain their score due to the unresolved fundamental issues. The positive reviewer (bad3) might maintain their score.

---

### Decision · Program_Chairs · 2026-01-26

Reject